# The establishment of variant surface glycoprotein monoallelic expression revealed by single-cell RNA-seq of *Trypanosoma brucei* in the tsetse fly salivary glands

Sebastian Hutchinson[1¤a]*, Sophie Foulon[2], Aline Crouzols[1], Roberta Menafra[2¤b], Brice Rotureau[1], Andrew D. Griffiths[2], Philippe Bastin[1]

1 Trypanosome Cell Biology Unit and INSERM U1201, Institut Pasteur, Paris, France, 2 Laboratoire de Biochimie, CBI, ESPCI Paris, Université PSL, CNRS UMR 8231, Paris, France

¤a Current address: Laboratoire de Biochimie, CBI, ESPCI Paris, Université PSL, CNRS UMR 8231, Paris, France
¤b Current address: Leiden Genome Technology Center, Department of Human Genetics, Leiden University Medical Center, Leiden, Netherlands
* sebastian.hutchinson@espci.fr

**Data Availability Statement:** All sequencing data generated in this study is available from the European nucleotide archive, accession number

## Abstract

The long and complex *Trypanosoma brucei* development in the tsetse fly vector culminates when parasites gain mammalian infectivity in the salivary glands. A key step in this process is the establishment of monoallelic variant surface glycoprotein (*VSG*) expression and the formation of the VSG coat. The establishment of VSG monoallelic expression is complex and poorly understood, due to the multiple parasite stages present in the salivary glands. Therefore, we sought to further our understanding of this phenomenon by performing single-cell RNA-sequencing (scRNA-seq) on these trypanosome populations. We were able to capture the developmental program of trypanosomes in the salivary glands, identifying populations of epimastigote, gamete, pre-metacyclic and metacyclic cells. Our results show that parasite metabolism is dramatically remodeled during development in the salivary glands, with a shift in transcript abundance from tricarboxylic acid metabolism to glycolytic metabolism. Analysis of *VSG* gene expression in pre-metacyclic and metacyclic cells revealed a dynamic *VSG* gene activation program. Strikingly, we found that pre-metacyclic cells contain transcripts from multiple *VSG* genes, which resolves to singular *VSG* gene expression in mature metacyclic cells. Single molecule RNA fluorescence *in situ* hybridisation (smRNA-FISH) of *VSG* gene expression following *in vitro* metacyclogenesis confirmed this finding. Our data demonstrate that multiple *VSG* genes are transcribed before a single gene is chosen. We propose a transcriptional race model governs the initiation of monoallelic expression.

## Author summary

African trypanosomes are parasitic protists which cause endemic disease in sub-Saharan Africa. To evade mammalian immune responses the parasite has developed a system of

PRJEB43345. All scripts, code and original microscopy images used to generate the figures are available from Zenodo 10.5281/zenodo.3974628.

**Funding:** This project has received funding from the European Union's (https://ec.europa.eu/) Horizon 2020 research and innovation programme under the Marie Skłodowska-Curie grant agreement No 794979 to S.H. S.H. was funded by a European Union Marie Skłodowska-Curie (No 794979) and an Institut Pasteur (https://www.pasteur.fr/) Roux-Cantarini fellowship. Funding was provided by the Institut Pasteur, and the French Government Agence Nationale de la Recherche (https://anr.fr/) Investissement d'Avenir Programme—Laboratoire d'Excellence "Integrative Biology of Emerging Infectious Diseases" (ANR-10-LABX-62-IBEID) to P.B. This work has received the support of "Institut Pierre-Gilles de Gennes" (laboratoire d'excellence, "Investissements d'avenir" program ANR-10-IDEX-0001-02 PSL, ANR-10-LABX-31 and ANR-10-EQPX-34. R.M. was supported by the Agence Nationale de la Recherche project "Cellectchip" ANR-14-CE10-0013 to A.D.G. ICGex Next Generation Sequencing platform of the Institut Curie supported by the grants ANR-10-EQPX-03 (Equipex) and ANR-10-INBS-09-08 (France Génomique Consortium) from the Agence Nationale de la Recherche ("Investissements d'Avenir" program), by the Canceropole Ile-de-France and by the Site de Recherche Intégrée sur le Cancer (https://siric.curie.fr/) - Curie program - SiRIC Grant INCa-DGOS- 4654. The funders had no role in study design, data collection and analysis, decision to publish, or preparation of the manuscript.

**Competing interests:** I have read the journal's policy and the authors of this manuscript have the following competing interests: A.D.G is a shareholder in 1CellBio.

antigenic variation, where the surface of the cell is covered in a tightly packed coat of variant surface glycoproteins (VSGs). Each cell expresses only one variant surface glycoprotein at a time, and this is periodically switched to evade new antibodies. The process of singular gene expression is termed monoallelic expression and this has two components, establishment and maintenance, i.e. how a single gene is selected for expression and how its singular expression is maintained throughout successive generations. The establishment of monoallelic *VSG* gene expression occurs in the salivary gland of the tsetse fly vector, although this process is not well understood. We used single cell gene expression profiling applied to thousands of single cells in the salivary gland of the fly. We show that in order to select a single gene, trypanosomes initially transcribe multiple *VSGs* before a single gene is selected for high-level expression. We propose a model where this process is driven by a race to accumulate transcription factors at a single *VSG* gene.

## Introduction

African trypanosomes are single-cell flagellated parasites that cause Human African Trypanosomiases, and nagana in cattle in sub-Saharan Africa. The parasites have a digenetic life cycle, cycling through a mammalian reservoir host and a tsetse fly vector host. During the developmental phase in the fly, the parasite progresses through multiple stages before acquiring mammalian infectivity in a process called metacyclogenesis in the salivary glands [1]. Trypanosome cells can be classified into multiple stages grouped in two morphotypes: trypomastigote and epimastigote forms (S1 Fig), defined by the relative positions of the nuclear and mitochondrial (kinetoplast) DNA within the cell [2]. Following a tsetse blood meal, the trypomastigote stumpy-form parasites ingested from the blood rapidly differentiate into trypomastigote procyclic form (PCF) parasites that will either directly colonise the cardia of the fly, or reach the posterior midgut before elongating and then migrating through the ectoperitrophic space back to the cardia (or proventriculus). In each scenario, only a small subset of cells eventually differentiates into epimastigote forms during a narrow time frame [3].

After a first asymmetric division, epimastigote parasites then migrate to the salivary glands where they attach to the epithelium via their flagellum and start proliferating to colonise it [4,5]. This population of attached epimastigote cells enters another asymmetric division, producing trypomastigote pre-metacyclic cells which then mature further to produce free mammalian-infective metacyclic cells that fill the lumen of the salivary glands and will be injected to the next host with the tsetse saliva [6] (S1 Fig). The two mitotic cell divisions of attached epimastigote cells, producing either an attached epimastigote or a pre-metacyclic trypomastigote daughter cell occur continuously and in parallel during the entire life of the tsetse, optimizing parasite transmission. An additional heterogeneous parasite population is found in the salivary glands, composed of meiotic cells and gametes [7,8]. These processes lead to a heterogeneous population of cells within the salivary glands. This heterogeneity is evident in previous scRNA-seq of salivary gland trypanosomes, where clusters of cells were identified corresponding to epimastigotes and metacyclic parasites [9]. RNA binding proteins appear to be the key regulators underpinning the trypanosome developmental program [10]. For instance, ALBA3 restricts development in the cardia [11] whereas overexpression of RNA binding protein 6 (RBP6) is sufficient to drive differentiation to infective metacyclic cells [12].

Differentiation of trypanosomes differentiating in different organs of the tsetse fly has been studied using bulk transcriptomics, revealing coordinated gene expression programs during development in the fly, for instance in the expression of RNA binding proteins, metabolite

transporters and surface protein transcripts [12,13]. By leveraging *in vitro* modelling of meta-cyclogenesis using inducible overexpression of RBP6, the proteomic and transcriptomic changes associated with metacyclogenesis are now understood. A combined approach of transcriptomics and proteomics showed changes in RNA binding proteins, surface proteins, clathrins, and energy metabolism, demonstrating the extent of the remodeling during metacyclogenesis [14]. Energy metabolism is well characterized in bloodstream form (BSF) and PCF trypanosomes. In the mammalian host, parasites utilize glucose as a primary carbon source to produce ATP through the glycolytic pathway that is restricted in specialized organelles named glycosomes. In contrast, in the tsetse fly midgut, parasites mostly use proline to feed the mitochondrial tricarboxylic acid (TCA) cycle and oxidative phosphorylation pathways to produce ATP (review in [15]). Finally, [16] used a combination of proteomics, transcriptomics and metabolomics together with detailed time-series analysis of RBP6 driven metacyclogenesis to delineate detailed changes in energy metabolism, cell remodeling and signaling pathways.

A key step in the metacyclogenesis process is the establishment of monoallelic *VSG* gene expression [17]. African trypanosomes have an exclusively extracellular lifecycle, meaning that in the mammalian host they are continually exposed to the immune system. To evade humoral immune attack, the parasite covers its surface in a monolayer of $10^7$ molecules of a single species of VSG protein which protects from complement mediated lysis [18,19]. It is put under continuous pressure to switch from the adaptive immune system, and the parasite evades this by antigenic variation, a process of VSG switching [20].

The epigenetic control of monoallelic expression of *VSG* genes underpins this antigenic variation [21]. Transcription of *VSG*s is highly structured; *VSG* genes are always transcribed from a telomere proximal site known as a *VSG*-expression site (*VSG*-ES), by RNA polymerase I (RNA Pol I) [22,23]. The genome has 19 bloodstream form *VSG*-ES which are polycistronic transcription units, and 8 monocistronic metacyclic *VSG*-ES (*mVSG*-ES) [24,25]. In addition, the trypanosome genome encodes several thousand silent *VSG* genes and gene fragments in a sub-telomeric archive which hugely increases the antigen diversity available for antigenic variation [26]. The single active *VSG*-ES is transcribed from a sub-nuclear organelle called the expression site body (ESB), an extra-nucleolar RNA Pol I focus [27]. Transcription is initiated at all *VSG*-ES, however only elongates over a single *VSG*-ES [28]. Maintenance of monoallelic *VSG* expression depends on the *VSG* Exclusion complex (VEX) complex, which associates with the ESB [29,30], and an mRNA trans-splicing locus, forming a transcription and splicing factory [31]. In addition, several chromatin factors, including the sheltrin component RAP1 and the histone chaperone CAF1 have been implicated in maintenance of monoallelic expression [30,32,33].

The establishment of monoallelic expression in metacyclic trypanosomes is relatively poorly understood. Immunogold staining of salivary gland trypanosomes and double immunofluorescence staining of *in vitro* derived metacyclic parasites demonstrates that monoallelic expression is established in the salivary glands of the tsetse fly [17,34]. In bloodstream parasites, monoallelic expression can undergo transcriptional switching events, where the active *VSG*-ES is switched off and a previously silent one is activated epigenetically [20], requiring the re-establishment of monoallelic expression. Analysis of BSF cells following a forced *VSG*-ES switch showed that transcription transiently increased at silent *VSG*-ES prior to switching the active *VSG*-ES. This was proposed as "probing" of silent *VSG*-ES before commitment to switching [35]. Only one factor has so-far been linked to the establishment of monoallelic expression; the histone methyltransferase DOT1b trimethylates Histone H3 lysine 76 [36] and is required to switch off a previously active *VSG*-ES following the establishment of a new active site [37].

The limiting nature and diverse cellular composition of salivary gland trypanosomes has made these parasites less experimentally tractable than *in vitro* systems. These parasites are not amenable to axenic culture (metacyclic cells are G0 arrested), and genetic manipulation tools are not as well established as in cultured parasites. We therefore applied a single-cell RNA-sequencing (scRNA-seq) approach to salivary gland derived trypanosomes, which overcomes many experimental boundaries through increased sensitivity and granularity to address the lack of resolution in these cell types. Here, we dissect the architecture of parasite development as they acquire mammalian infectivity in the salivary glands using inDrop. InDrop is a scRNA-seq technology wherein single cells are co-encapsulated inside nanoliter droplets with single barcoded hydrogel microspheres (BHMs) coupled to barcoded cDNA primers that hybridize to the 3' poly(A) tail of mRNA, allowing sequencing of several thousand cells per experiment. Reverse transcription of mRNAs from each cell generates cDNAs tagged with a unique barcode, specific to each cell, permitting the informatic decomposition of single-cell transcriptomes [38,39].

In this study, we sought to understand how monoallelic VSG expression is established in trypanosomes. We first demonstrated that trypanosomes are compatible with 3′ mRNA barcoding using inDrop BHMs. After having generated single cell transcriptomes for cultured bloodstream (BSF) and PCF trypanosomes, we performed inDrop with parasites from the salivary glands. This revealed the presence of 4 cell clusters in the salivary glands, corresponding to 1) attached epimastigote cells, 2) gametes and meiotic cells, 3) pre-metacyclic trypomastigote cells and 4) metacyclic parasites. We delineated transcriptomic changes to energy metabolism pathways, revealing a coherent developmental program in this organ towards infectivity. Finally, we have performed the first analysis of the establishment of monoallelic VSG expression *in vivo*. We show that the establishment of monoallelic expression is a two-step process, where pre-metacyclic cells first initiate expression of multiple *mVSG* transcripts, followed by monoallelic expression in metacyclic parasites. We further validated this finding by an orthogonal method, employing smRNA-FISH with parasites undergoing metacyclogenesis *in vitro*.

## Results

### Assessment of single-cell RNA-seq using inDrop applied to *Trypanosoma brucei*

To test whether trypanosomes would not be visibly stressed by the inDrop pipeline, we monitored the survival and RNA metabolism using ALBA3 and DHH1 as markers for cellular stress in procyclic form (PCF) cells, in conditions which mimic the inDrop protocol. Following starvation in PBS for 2 h at 27°C, ALBA3 and DHH1 form RNA granules in the cytoplasm [11], however when cells are placed on ice for 2h in phosphate buffered saline, no RNA granule formation was observed (S2A Fig). In addition, cells remained alive based on their motility after returning them to 27°C SDM-79 medium (97% untreated and 94% 2h PBS) (S2B Fig), indicating that they are minimally stressed in the conditions required for inDrop.

We first characterized the performance of the inDrop pipeline with cultured Antat1.1E PCF trypanosomes expressing EP (Glu-Pro) / GPEET (Gly-Pro-Glu-Glu-Thr) repeat containing procyclins. *T. brucei* mRNAs are poly-adenylated and therefore compatible with priming of cDNA synthesis using poly(T)VN, so we began by performing 'bulk' experiments using a primer containing a T7 promoter for cDNA amplification by *in vitro* transcription (IVT), an Illumina sequencing adaptor, a single inDrop cell barcode, unique molecular identifiers UMI [40] and a poly(T)VN sequence for mRNA capture (see Materials and methods). Library sequencing revealed that the majority of barcoded cDNAs corresponded to the annotated 3′-end of transcripts, as expected (Fig 1A). We therefore proceeded to single cell barcoding of

trypanosomes. We mixed cultured BSF and PCF trypanosomes (47.5% each) with Ramos cells (5%), a human B-cell line derived from Burkitt's lymphoma [41] as a positive control. The inclusion of Ramos cells also allows us to estimate the amount of ambient RNA within the cell suspension which is detected by inDrop. We constructed a sequencing library, and generated 77.4 million reads. To facilitate the use of trypanosome genomic sequences for which no reference strain is available, we developed a bespoke analysis pipeline to create a count matrix (see Materials and methods). After processing the reads with this pipeline, we used the Seurat and single cell transform (SCT) R packages for data analysis [42–44] and determined that the transcriptomes of 1,979 cells were captured, with an average of 2,195 UMIs and 1,347 genes per cell. The emulsion contained approximately 4,900 cells, suggesting that we recovered approximately 40% of the cells through inDrop. Dimensionality reduction using the Uniform Manifold Approximation and Projection (UMAP) algorithm [45] revealed three clusters which corresponded to the three expected cell types in approximately the expected proportions (Fig 1B), with 164 (8%) Ramos cells, 809 (41%) procyclic cells and 1,006 (51%) BSF cells. Within clusters, we detected an average of 1,932 UMIs and 1,211 genes for BSF, 2,268 UMIs and 1,258 genes for PCF cells and an average of 3,447 UMIs and 2,627 genes in Ramos cells (Fig 1C). Each trypanosome cell has between 20,000–40,000 mRNA molecules per cell [46] therefore these data are consistent with an mRNA capture efficiency between 4.8% and 11.3%, similar to the 7.1% reported by [38]. We next plotted percentage of human and trypanosome UMIs for each barcode (S2C Fig). This revealed that human barcodes contained higher proportions of trypanosome UMIs than the converse. We hypothesized that this was potentially due to trypanosome cell fragility which leads to release of trypanosome mRNAs from lysed cells. We therefore modified our protocol to maintain cell suspensions on ice until the moment they are encapsulated (see Materials and methods).

Marker gene analysis for each cluster revealed significantly differentially expressed genes, corresponding to the known biology of each respective cell type. In the BSF cluster, marker genes included surface proteins such as the active *VSG-2* (Tb427.BES40.22) (Fig 1D), *GRESAG 4* (Tb927.7.6080), *ISG65* (Tb927.2.3270), and *GPI-PLC* (Tb927.2.6000), as well as transcripts involved in the glycolytic pathway, such as *AOX* (Tb927.10.7090), *ALD* (Tb927.10.5620), and *PYK1* (Tb927.10.14140), for example. The PCF cluster markers included the procyclic surface markers *EP1* (Tb927.10.10260) (Fig 1D), *EP2* (Tb927.10.10250), *GPEET* (Tb927.6.510) and *PARP A/EP3-2* (Tb927.6.450), the electron transport chain component *COXIV* (Tb927.1.4100), an amino acid transporter (*AAT10-2*) (Tb927.8.8300), the RNA binding protein *UBP2* (Tb927.11.510) and *HSP90* (Tb927.10.10980), all of which are seen to be up-regulated in PCF using bulk RNA-seq [47]. In the Ramos cluster, markers included genes such as immunoglobulin lambda constant 3 (*IGLC3*) (Fig 1D) and prothymyosin alpha (*PTMA*), as expected. These data correspond with known biology from bulk transcriptomics experiments and therefore demonstrate that inDrop is capable of single-cell analysis of trypanosome transcripts.

## Single-cell analysis of trypanosome salivary gland transcriptomes using inDrop

We next proceeded to single-cell barcoding of trypanosomes from tsetse fly salivary glands. We infected cages of 50 teneral (1–2 days post-eclosion) male tsetse flies with *in vitro* generated stumpy-form Antat1.1E (EATRO1125) *T. b. brucei* and recovered salivary gland parasites by dissection 28 days post infection. We obtained 4.6 x10$^4$ parasites from a total of 72 dissected flies (S3 Fig). We then generated and sequenced single-cell barcoded libraries (S4A Fig), and

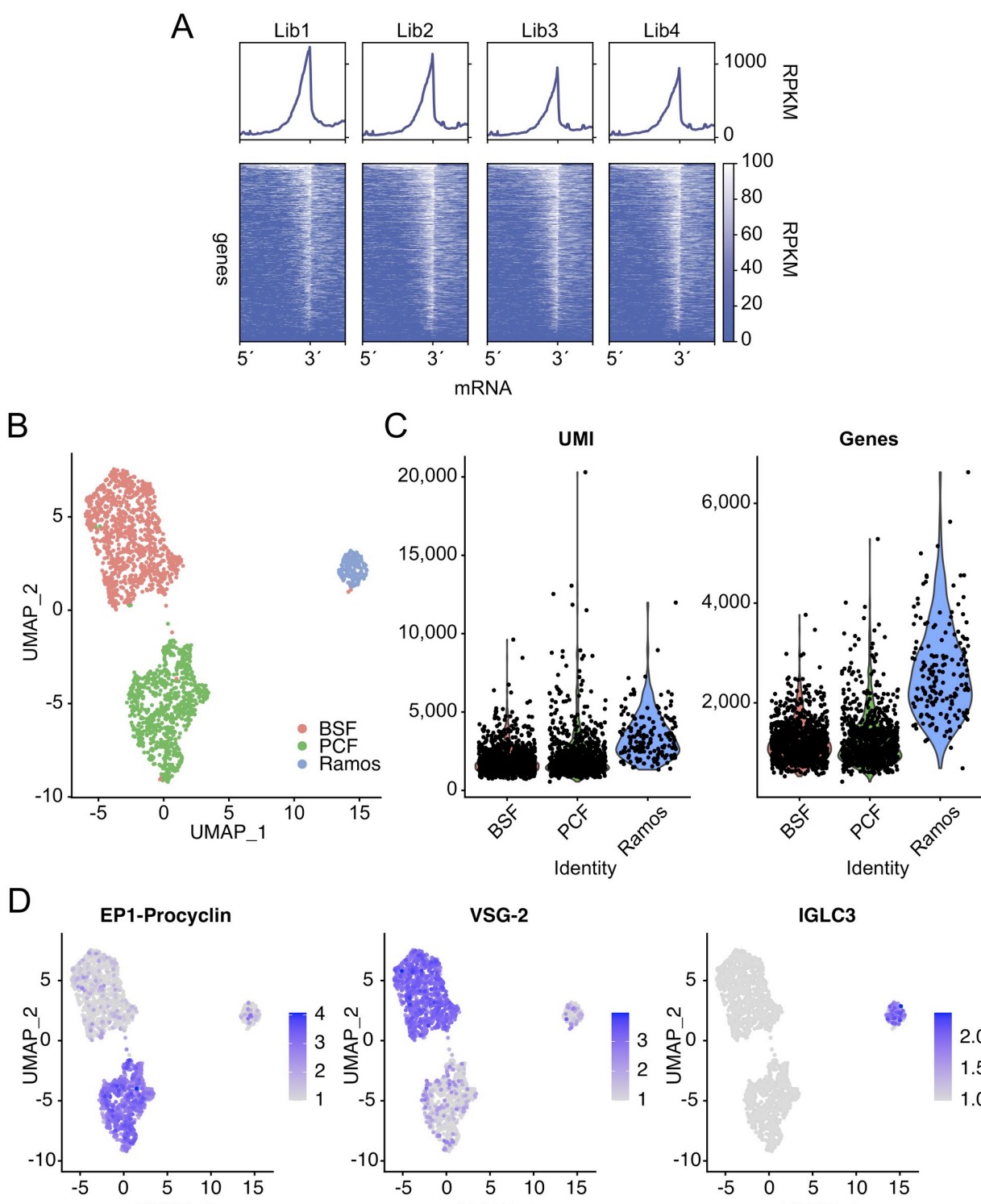

**Fig 1. inDrop analysis of cultured bloodstream form and procyclic trypanosomes.** A. inDrop performed in a microfluidic format, using a single primer to replace the BHMs (no single-cell resolution). The plots show the reads per kilobase per million mapped (RPKM) for annotated *T. brucei* transcripts aligned from their 5′splice site to the 3′polyadenylation sequence. Profile plots (above) show the average RPKM across the ~7,500 non-redundant gene set for trypanosomes. Heatmaps (below) show the RPKM for individual transcripts. C. UMAP projection of single-cell barcoding data for a population of cells containing calculated proportions of 47.5% for each BSF and PCF, and 5% Ramos cells. Measured cell proportions are 51% BSF, 41% PCF and 8% Ramos. D. Violin plots depict the total number of UMIs and genes captured per cell within BSF, PCF and Ramos cell clusters. E. UMAP projections from C overlaid with colour scale of gene expression values for *EP1* procyclin (Tb927.10.10260), *VSG-2* (Tb427.BES40.22) and Immunoglobulin lambda constant 3 (IGLC3).

developed an optimized bioinformatic pipeline for *T. brucei*, recovering data for 2,279 cells from two technical replicates, detecting an average of 959 UMIs per replicate (Fig 2 and S4B Fig). As no genome sequence is available for the Antat1.1 strain used here, we identified *VSG* genes expressed in the salivary glands by the Antat1.1 strain by aligning reads to the *VSGnome* for Antat1.1E (EATRO1125) [48] (S5 Fig). We then selected the top 8 sequences, in line with the reported number of *mVSG*-ES in the Lister-427 strain [25]. We were able to identify linked *mVSG*-ES promoters for 5 of these genes (VSG-4959, VSG-1654, VSG-4862, VSG-4564 and VSG-385) to metacyclic expression site promoters using BLAST [49] to identify similar sequences up-stream [48,50].

We used the Seurat R package to analyze our technical replicates [43,44]. Count matrices were compiled from all aligned reads, except for *VSG* genes, where we used a stringent mapping quality (MapQ40), to avoid mis-aligned reads being spuriously counted. We next examined several technical aspects of inDrop, Firstly, we examined the background from ambient RNA by looking at the detection rate in a cluster of Ramos cells, which were incorporated into our experiment. These data showed that we did not detect more than 2 *mVSG* UMIs per barcode in Ramos cells (S6A Fig). We therefore applied a >2 UMI cutoff for our *mVSG* analyses below. We established the percentage of stringently aligned reads (mapping quality >40) in our data. This showed improvement over our *in vitro* data, with two distinct clusters of cells for human and trypanosome UMIs (S6B Fig). We next considered whether sequencing errors could contribute to barcode switching. Our bioinformatics pipeline permits up-to two errors per cell barcode (cell barcode1 is between 8 and 11 nt and cell barcode2 is 8 nt [39]), so we compared the pairwise Levenshtein distance (the minimum number of single character edits required to permute one barcode to the other) between all cell barcodes in our study. The mean Levenshtein distance between barcodes was 11.4 nt, and only 0.017% of barcode pairs were within a Levenshtein distance of 2 (S6C Fig). These parameters allowed us to effectively estimate the noise for our downstream analyses.

Following data preprocessing and integration of the technical replicates, we visualized the datasets using UMAP projections [45] (Fig 2A). Clustering identified 5 clusters (Fig 2B), whose developmental state could be identified using known marker genes (Fig 1D). *EP1 Procyclin* (Tb927.10.10260) is a surface protein found in midgut forms [51], and its transcript can also be detected in salivary gland parasites [52]. Brucei alanine-rich protein (*BARP*, Tb927.9.15640) is a surface protein present in attached epimastigote parasites [52]. Metacyclic VSG expression is a marker for metacyclic cells [6], and *VSG-393* is the most abundant *VSG* transcript expressed in our dataset. RBP6 (Tb927.3.2930) is an RNA binding protein which drives developmental progression to infectivity in salivary glands, and is found in epimastigote forms in the salivary glands [9,12]. Calflagin is a flagellar calcium binding protein whose expression is up-regulated in pre-metacyclic and metacyclic forms [6]. Finally, HAP2 (Tb927.10.10770) is a membrane-membrane fusion protein found conserved in nature and expressed in the gametes of many species (Fig 2C) [53]. Further evidence from bulk transcriptomics of midgut, cardia (proventriculus) or salivary gland tissues [12] were used to identify midgut forms. Of the 226 marker genes identified for the midgut forms cluster in our inDrop data, 82.3% of genes were maximally expressed in the midgut samples of bulk transcriptomics [12], further supporting this annotation. We believe the presence of these cells is likely due to contamination during fly dissection. In total, we identified 142 midgut forms, 296 attached epimastigote cells, 280 gametes, 317 pre-metacyclic and 1,244 metacyclic cells. These data indicate that we are able to capture single cell transcriptomes of these salivary gland parasites, and identify cell states from these transcriptomes, using inDrop.

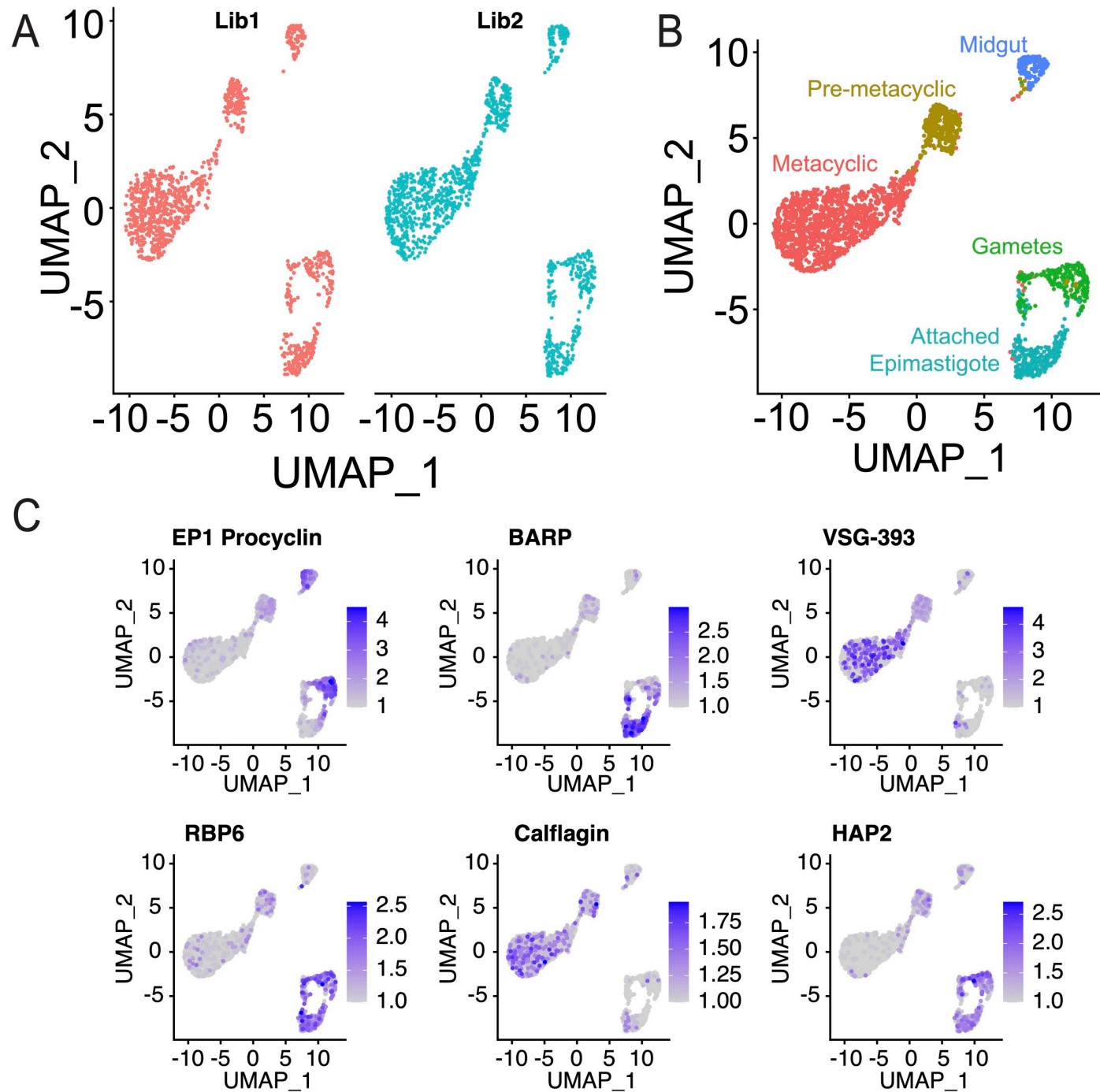

**Fig 2. InDrop barcoding of salivary gland _T. brucei_.** A. UMAP projection of two technical replicates (Lib1 and Lib2, red and blue, respectively). B. Merged UMAP projection of technical replicates with clustering information as indicated. C. Gene expression levels for marker genes in the salivary glands, _EP1 procyclin_ (midgut stages) [51]: Tb927.10.10260, _BARP_ (epimastigote stage): Tb927.9.15640 [52], _VSG-393_: (Genbank) KC612418.1 [48], _RBP6_: Tb927.3.2930 [12], _Calflagin_: Tb927.8.5440 [6], _HAP2_: Tb927.10.10770 [53].

## Metabolic transcript analysis reveals dramatic remodeling during metacyclogenesis

To further our understanding of developmental changes in the salivary glands, we assessed the expression of glycolysis and TCA cycle genes expressed in the transcriptomes of parasites from

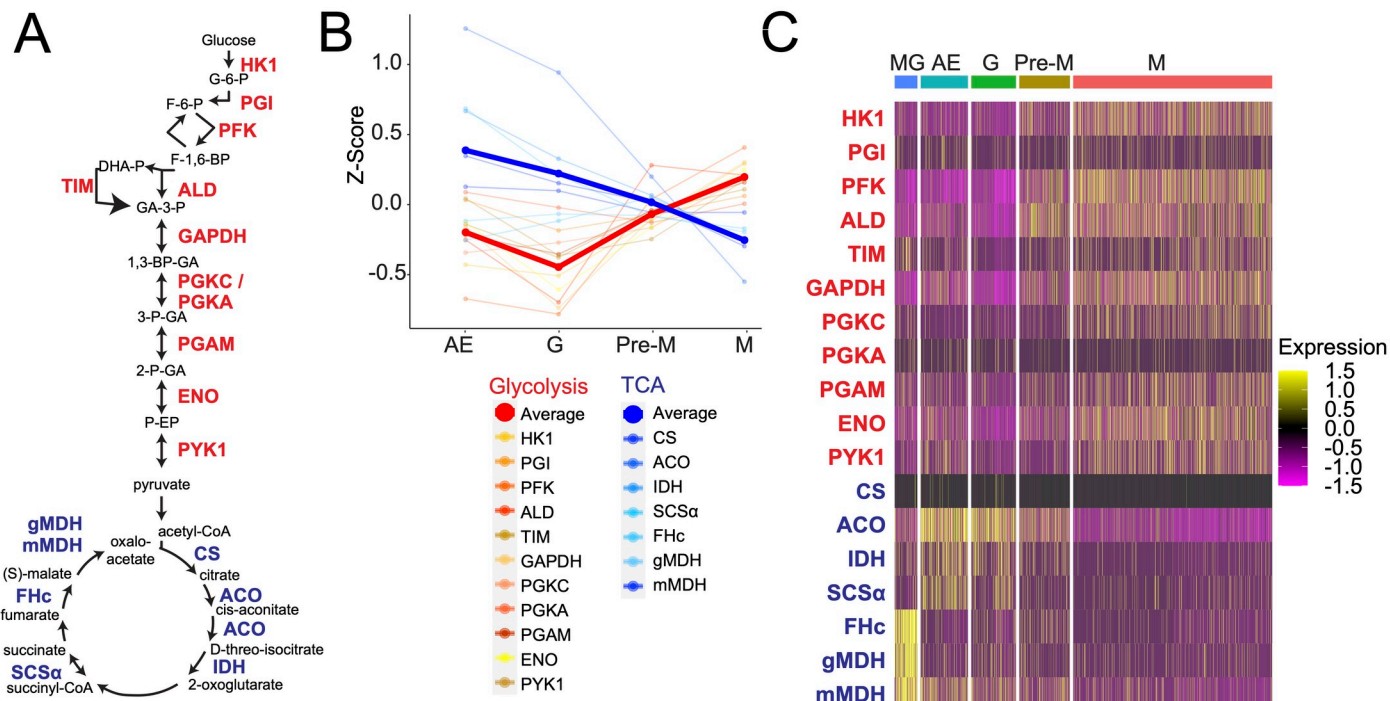

**Fig 3. Cluster-based analysis of main energy metabolism transcripts.** A. The schematic shows the glycolytic and TCA pathways [82]. Compounds (black text) and enzymes (red text for glycolysis, blue text for TCA cycle) are shown and reactions are represented by arrows. B. Plot shows the expression Z-scores for each glycolysis or TCA cycle enzyme transcript retained after SCT normalization [43]. Thicker lines show the average across each cohort. AE: attached epimastigote cells; G: Gamete cells; Pre-M: pre-metacyclic cells: M: metacyclic cells. C. Single cell level analysis of metabolic transcripts. Cells are arranged by cluster (left to right) and relative transcript expression shown per cell. Colours represent Z-transformed expression values. Cluster abbreviations as for B. Abbreviations of metabolites: G-6-P: glucose-6-phosphate; F-6-P: fructose-6-phosphate; F-1,6-BP: fructose-1,6-bisphosphate; DHA-P: dihydroxyacetone phosphate; GA-3-P: glyceraldehyde-3-phosphate; 1,3-BP-GA: 1,3-bisphosphoglycerate; 3-P-GA: 3-phosphoglycerate; 2-P-GA: 2-phosphoglycerate; P-EP: phospho-enol pyruvate. Abbreviations of enzymes: HK1: hexokinase 1; PGI phosphoglucose isomerase; PFK: phosphofructokinase; ALD: aldolase; TIM: triose-phosphate isomerase; GAPDH: glyceraldehyde-3-phosphate dehydrogenase; PGK(C/A): phosphoglycerate kinase, PGAM: phosphoglycerate mutase; ENO: enolase; PYK1: pyruvate kinase; CS: citrate synthase; ACO: aconitase; IDH: isocitrate dehydrogenase; SCSα: succinyl coenzyme A synthetase; FHc, fumarate hydratase, cytosolic; gMDH: glycosomal malate dehydrogenase; mMDH: mitochondrial malate dehydrogenase.

the salivary glands. Our analysis reveals that parasites remodel the expression of metabolic enzymes during differentiation to metacyclic cells (Fig 3 and S7 Fig). We were able to assess the expression changes for 11 glycolysis enzyme transcripts and 7 TCA cycle transcripts (Fig 3A). We observed a common expression profile for most genes at the cluster (Fig 3B and S7 Fig) and single cell level (Fig 3C). Glycolytic enzymes increased in their expression between pre-metacyclic and metacyclic clusters, with hexokinase (*HK1*, Tb927.10.2010) and ATP-dependent 6-phosphofructokinase (*PFK*, Tb927.3.3270) being clear examples of this. We observed a decline in TCA cycle enzyme expression during metacyclogenesis, with pre-meta-cyclic cells positioned as a clear intermediate stage between epimastigote forms (Attached and Gametes) and metacyclic cells. For instance, aconitase (*ACO*, Tb927.10.14000) and glycosomal isocitrate dehydrogenase (*IDH*, Tb927.11.900) both decrease from their peak expression in attached epimastigote cells to their lowest in metacyclic parasites with pre-metacyclic cells as intermediates. This indicates that the data clustering is indeed reflecting the true developmental progression of these cells, consistent both with previous electron microscopy observations on glycosomal and mitochondrial remodeling during this development [12,54], and previous multi-omic evidence from these cell types [9,16].

## Identification of a putative gamete cluster

A subpopulation of trypanosomes undergoes meiosis I and II in the salivary glands, producing gamete cells that can recombine sexually [8]. *HAP2* expression has not been previously profiled during parasite development in the salivary glands. In our data, expression of *HAP2* was significantly upregulated in gamete cells compared to the other clusters (average log2 FC 0.74, detected in 82.5% gamete cells and 36.6% cells in other clusters, adjusted p.val = 0. 9.75 x $10^{-50}$, negative binomial test, S1 Table). Additionally, and unexpectedly, *BARP* expression was significantly reduced in gamete cells compared to attached epimastigote cells (average log2 FC -0.83, detected in 94.6% attached epimastigote cells and 61.1% gamete cells, adjusted p. val = 9.68 x $10^{-36}$, negative binomial test) and *EP1 procyclin* expression up-regulated (average log2 FC 1.45, detected in 58.1% and 87.5% of attached epimastigote cells and gamete cells, respectively, adjusted p.val = 3.26 x $10^{-56}$), potentially explaining the presence of these transcripts in previous studies [52]. We also detected a significant up-regulation of the gametocytogenesis marker *HOP1* (Tb927.10.5490) [8] in this cluster compared to the other clusters (average log2 FC 0.80, detected in 7.5% gamete cells and 1.8% cells in other clusters, adjusted p.val = 1.54 x $10^{-4}$). These data therefore support the assignment of gamete cells to this cluster.

Attached epimastigote cells undergo asymmetrical division to produce one attached epimastigote daughter and one pre-metacyclic trypomastigote daughter [6]. We examined these cells in more detail (Fig 4A). We searched for cells expressing the histones (*H1* Tb927.11.1800, *H2A* Tb927.7.2820, *H2B* Tb927.10.10460, *H3* Tb927.1.2430 and *H4* Tb927.5.4170), as a marker for S-phase cells [55,56], and discovered two sub-clusters of cells expressing these transcripts (Fig 4B). To investigate this more quantitatively, we sub-divided the S-Phase cluster and assessed the marker genes expressed in these clusters (Fig 4C). This revealed that *BARP* and the set of 5 canonical histones were significantly associated with the "BARP-S" cluster. *HAP2* was significantly associated with both clusters, albeit more strongly with the HAP2-S cluster (S2 Table). The presence of these two clusters raises the questions as to whether there are two potential modes of development in the salivary glands.

## Establishment of *mVSG* monoallelic expression

VSG protein expression is initiated during pre-metacyclic to metacyclic differentiation [6,54]. Consistent with this, our data revealed, as expected, the presence of very few *mVSG* UMIs in attached epimastigote cells with 85% of cells containing no detectable *mVSG* transcripts, compared to pre-metacyclic and metacyclic clusters (S8 Fig). Using our Ramos cell data to estimate the amount of ambient RNA in the cell suspension detected by inDrop, we found that attached epimastigote cells contained more *mVSG* UMIs on average than Ramos cells, where these genes are not present (3.36 vs 0.19 UMIs per cell, respectively), suggesting that there is some transcription of these loci, consistent with ubiquitous transcription initiation from RNA-Pol I promoters [28]. We observed an increase in the average number of *mVSG* UMIs per cell as parasites proceeded through their developmental program (S8 Fig), increasing from a mean of 3.36 UMIs in attached epimastigote cells to 10.0 UMIs and 21.7 UMIs in pre-metacyclic and metacyclic cells, respectively.

We next assessed the diversity of *mVSG* expression during the differentiation process. Plotting the expression profile of individual pre-metacyclic cells showed that the majority of cells in this cluster express multiple different *mVSG* genes (Fig 5A). We quantified the number of different *mVSG* genes expressed per cell (i.e. the diversity of UMIs per cell) in cells with more than 10 *mVSG* UMIs. This revealed, strikingly, that pre-metacyclic cells expressed up-to six different *mVSG* genes. Multi-*mVSG* gene expression (UMI cutoff >2) was observed in 86.7% of cells, whereas only 13.3% expressed a single *mVSG* gene. (Fig 5B). Following differentiation

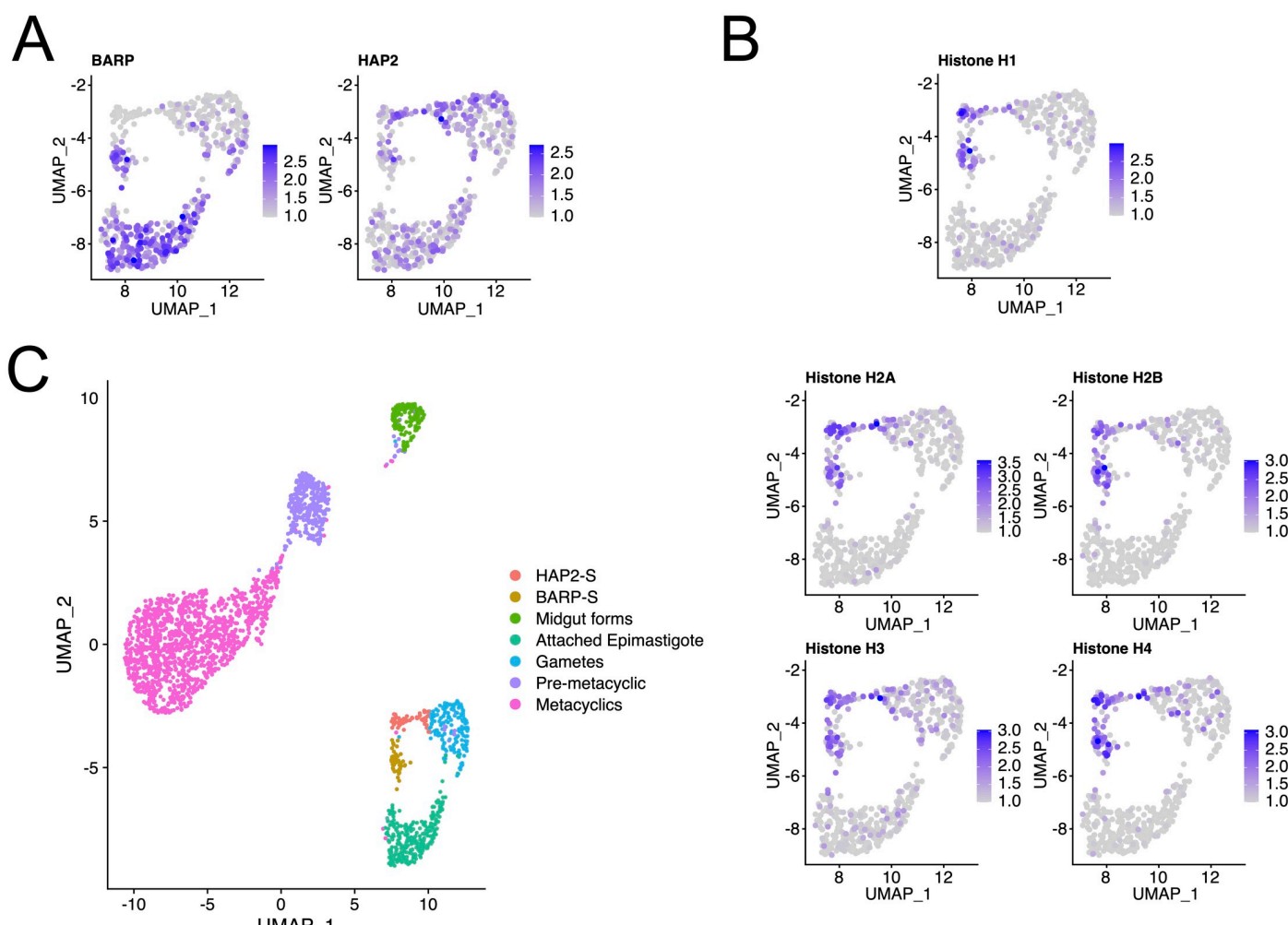

**Fig 4. Analysis of gamete cluster.** A. Normalised expression values for *BARP* (Tb927.9.15640) and *HAP2* (Tb927.10.10770) overlaid on UMAP projections of attached epimastigote and gamete cell clusters. B. Normalised expression values for canonical histones: H1 Tb927.11.1800, H2A Tb927.7.2820, H2B Tb927.10.10460, H3 Tb927.1.2430 and H4 Tb927.5.4170. C. Sub-clustering of cells in S-Phase. Clusters were manually re-annotated using the Seurat function CellSelector. Two new clusters of cells expressing both core histones and either BARP or HAP2 were created and are indicated in brown and red, respectively.

to metacyclic form cells, however, *mVSG* expression had resolved into a monoallelic state (Fig 5C), where a single *mVSG* gene was detected in 86.8% of cells, 12.2% contained two *mVSG* genes and only 1% expressed 3 genes types (Fig 5D). This transition in mVSG expression is also evidenced when overlaying the number of *mVSG* genes detected onto a UMAP projection (Fig 5E).

We queried whether the detection of multiple *mVSG* genes in pre-metacyclic cells was a function of sequencing depth for those barcodes, compared to metacyclic cells. We plotted the number of *VSG* genes detected per barcode against the number of aligned reads, stratified by cluster (S9 Fig). This revealed that rather than pre-metacyclic cells having higher aligned read counts than metacyclic cells, the median aligned reads per barcode associated to pre-metacyclic cells was lower than metacyclic cells. This was true even when comparing metacyclic cells with a single expressed *mVSG* to pre-metacyclic cells expressing 6 *mVSG* genes ($2.2 \times 10^4$ vs $1.6 \times 10^4$ median aligned reads, respectively). These data show that it is highly unlikely that the detection of multiple *mVSG* transcripts per cell is artefactual due to sequencing depth.

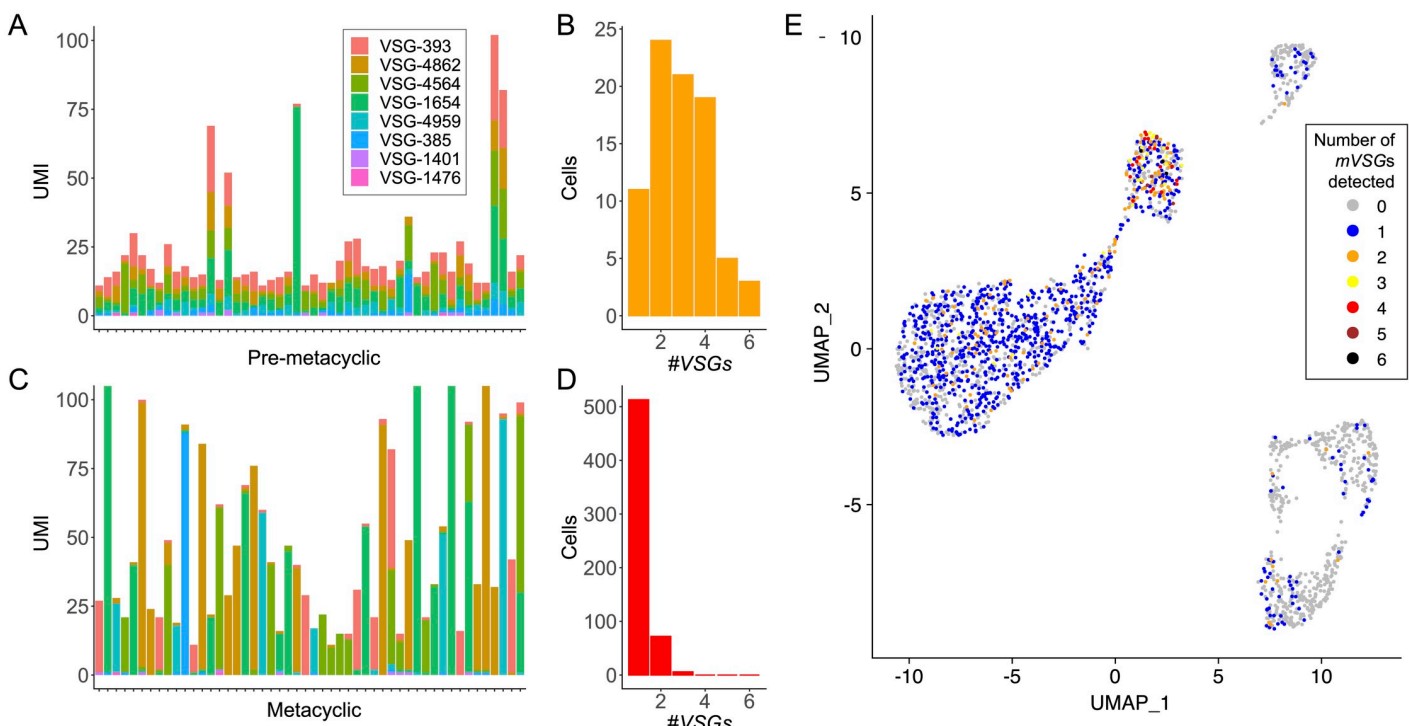

**Fig 5. Developmental progression of *VSG* expression.** A. *VSG* expression data for a random subset of 50 pre-metacyclic cells, with a total *VSG* UMI count > 10. Data are raw (unscaled) UMI counts. Each column represents a cell and each colour a different *VSG*. B. Histogram showing the number of *VSG* expressed for all pre-metacyclic cells (per *VSG* UMI count > 2, all cells in cluster). C. As for A except that the data are a subset of 50 metacyclic cells with total *VSG* UMI counts >10. D. Histogram as for B except showing metacyclic *VSG* transcript diversity (per *VSG* UMI count >2, all cells in cluster). E. UMAP projection from Fig 2B coloured according to the number of *mVSG* genes detected per cell.

We sought to further validate our findings by turning to previously published scRNA-seq data of salivary gland trypanosomes using both a different trypanosome strain; RUMP503, and a different technique, the 10xGenomics single-cell RNA-seq system [9]. As the complete set of *mVSG* genes was not available for this strain, we used Trinity [57] to assemble *mVSG* transcripts *de novo* from bulk RNA-seq data [13], and retained the 8 most abundant *mVSG* genes for downstream analysis (S10 Fig). This allowed us to perform a detailed analysis of *mVSG* expression in these data, which was not possible previously. We reprocessed the scRNA-seq data set using our pipeline, observing similar clusters to those published previously [9] (S11A and S11B Fig). We identified 6 clusters, with two each corresponding to attached epimastigote, intermediate and metacyclic cells, following published nomenclature [9] (S11A and S11B Fig). We next interrogated the *mVSG* expression profiles in each cluster, applying the same per-gene UMI cutoff as for our inDrop analysis (>2), revealing a developmental program (S11C and S11D Fig) where many *mVSGs* are detected in intermediate clusters (up to 7 genes). In epimastigote clusters 1 and 2, the majority of cells (76.8 and 66.7%, respectively) do not express detectable *mVSG* transcripts. In intermediate clusters 1 and 2, *mVSG* expression is initiated, with 64.8 and 8.2% of cells with no detectable *mVSG* expression, and with 17.6 and 44.7% of cells here containing two or more *mVSG* transcripts, the remainder with one. In the metacyclic cell clusters 1 and 2, a single *mVSG* gene was detected in 78.5 and 82.3% of cells, with only 8.3 and 7.7% of cells containing two or more *mVSG* genes (S11D Fig). This new finding is similar to our inDrop data, where 87% of pre-metacyclic cells expressed more than a single *mVSG*.

These new analyses uncover that expression of multiple *mVSGs* is a common precursor to monoallelic expression in different trypanosome strains *in vivo*.

To formally demonstrate the existence of cells expressing more than a single *VSG* gene, we performed smRNA-FISH following *in vitro* metacyclogenesis by overexpressing RBP6. This *in vitro* process models the establishment of monoallelic mVSG expression in a third strain, the Lister 427 genetic background. Induction of RBP6 in PCF forms triggers differentiation first to epimastigote forms, which then further differentiate to trypomastigote forms, producing infective metacyclic forms which express a single mVSG [12,34]. This process therefore allows us to model the initiation of *mVSG* expression.

We overexpressed RBP6 in PCF cells using a tetracycline inducible system [58], and sampled cultures after 3, 4 or 5 days of induction. After probing for *VSG-397*, *VSG-531* (the most abundant *mVSGs* observed by [12]) and alpha tubulin as a positive control, we detected *VSG* transcripts in ~5% of cells (303 cells positive for *VSG* out of a total of 6,452). Within this population, we established 3 categories of cells: "VSG+", where transcript numbers were quantifiable (193 cells), "VSG++" where transcript numbers were too high to be quantified accurately (but where individual transcripts were discernable) (18 cells), and "VSG+++", where individual transcripts were no longer discernable (92 cells) (for representative images see Fig 6A, S3 Table).

To visualize the proportions of *VSG-397* and *VSG-531* transcripts per cell, we estimated the total number of *mVSG* transcripts per cell in VSG++ and VSG+++ categories. BSF cells contain on average 2,000 *VSG* transcripts [46]. Hence, we set the number of transcripts in VSG+++ cells at 2,000, and VSG++ cells at 200 (an order of magnitude between VSG+ and VSG++) per channel (i.e. if both *VSG-397* and *VSG-531* were "VSG++", we estimate that cell contains 400 *VSG* transcripts). Cells were therefore classified as VSG+ with less than 200 transcripts, VSG++ with between 200 and 400 transcripts, and VSG+++ with 2000 transcripts or more. We then assessed singular VSG gene expression in each cell category: 30% (58 cells) VSG+ cells contained transcripts from both *VSG-397* and *VSG-531*, directly demonstrating the presence of cells expressing two different *VSG* genes. This proportion dropped to only 17% (3 cells) among VSG++ cells and 0% in VSG+++ cells, indicating that monoallelic expression is established as cells express a larger amount of *mVSG* transcripts (Fig 6B).

## Discussion

Our data demonstrate that the inDrop scRNA-seq approach is compatible with investigation of gene expression in trypanosomes. Despite significant biological constraints (limited access to multiple populations of parasites in low numbers), we were able to delineate the developmental progression of these parasites in the salivary glands, including the identification of a gamete cluster. In our hands, we detected similar numbers of trypanosome UMIs (~2,000 for cultured cells) compared to Ramos cells (~3,500), despite having a much lower mRNA content per cell ($10^5$ vs $10^4$ for human and trypanosome cells, respectively) [46]. The addition of Ramos cells was particularly useful in our experiments in establishing contribution of ambient RNA to noise in our analysis of the establishment of monoallelic *mVSG* expression. Although a technical comparison between inDrop and 10xGenomics profiling of single trypanosome transcriptomes is not possible (for a detailed technical comparison of the two technologies see [59]), our data are broadly comparable with previously published data [9]. We observe a similar profile of epimastigotes differentiating into metacyclics, and similar transcriptomic changes, e.g. expression of surface markers *BARP*, *VSG* and the RNA binding protein *RBP6* between clusters, for example.

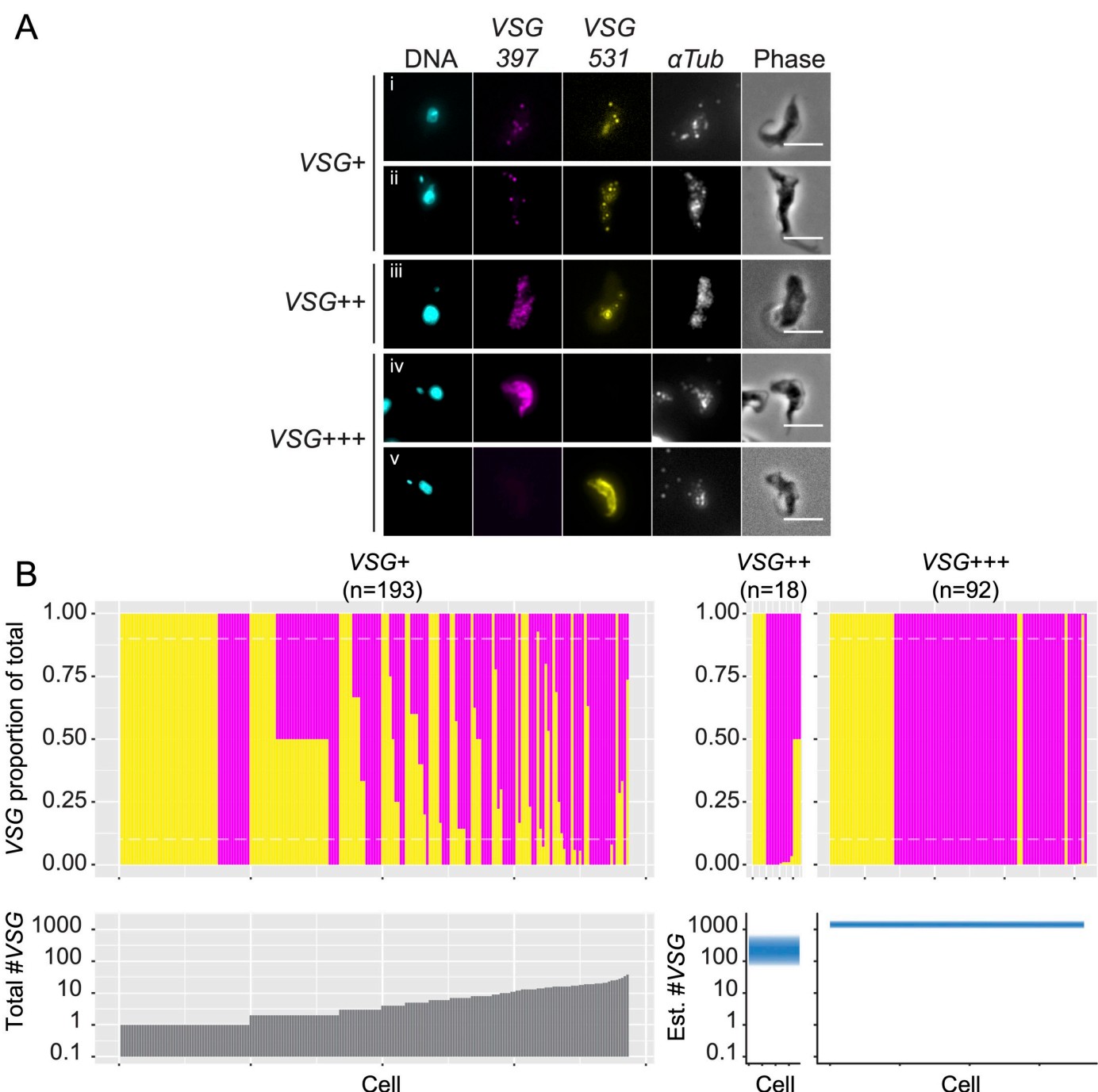

**Fig 6. Single molecule RNA-FISH of *in vitro* metacyclogenesis.** A. Panel of representative images of VSG+ and VSG++ cells containing both *VSG-397* and *VSG-531* transcripts (*i-iii*) and VSG+++ cells where a single VSG type is present (*iv* and *v*). B. Above, proportions of *VSG-397* and *VSG-531* transcripts for VSG+, VSG++ and VSG+++ cells. Each cell is plotted as a single column on the x-axis. An estimate of transcript abundance for VSG++ and VSG+++ cells was used based on published estimates of *VSG* transcript abundance in BSF [46]. Below, transcript abundance for VSG+ cells, and estimated transcript abundance for VSG+ and VSG++ cells.

The gamete cell cluster was characterized by a significant upregulation of *HAP2*, a conserved membrane fusion protein required for gamete fusion in diverse organisms [53], and *HOP1*, a component of synaptonemal complexes formed in the first meiotic division [8]. We did not, however, observe statistically significant changes in other known marker genes of

gametocytogenesis such as *SPO11*, *DMC1* or *MND1* [8], likely due the fact that inDrop only captures approximately 10% of cellular mRNAs [38]. Our observations of increased EP1 procyclin (Tb927.10.10260) transcripts in these cells is consistent with previous observations showing the presence of both procyclin transcripts and proteins in salivary gland parasites, although a detailed characterization of the procyclin expression in these cell types is currently lacking [52,60]. There are several possible reasons for our observation of a gamete cell cluster that was not observed previously [9]. In our experience, trypanosome infections of tsetse flies are highly variable, with factors such as the duration of infection and time since last feeding affecting the relative proportions observed cell types. In addition, the *HAP2* gene currently does not have an annotated 3′ UTR, therefore our 3′ extended transcript annotations likely captured more *HAP2* transcripts in this study.

Our analysis of the metabolic transcriptome remodeling indicated that energy metabolism in attached epimastigote cells is dependent on the production of ATP in the mitochondrion, consistent with recent evidence from *in vitro* derived epimastigote parasites using the RBP6 overexpression system [16]. The most dramatic transcriptome remodeling was following asymmetrical division of the attached epimastigote, with pre-metacyclic cells as transition intermediates in terms of metabolic remodeling (Fig 3). These data are consistent with electron microscopy evidence showing an increase in the number and size of glycosomes together with the reduction of mitochondrial branching in metacyclic cells compared to attached epimastigote parasites [12,54].

Our analysis of the establishment of monoallelic *mVSG* expression has uncovered that this is a phased process, first involving the activation of multiple *mVSG*-ES, and second the emergence of a single active *mVSG*-ES. Here, we present three independent lines of evidence to support this finding. First, in AnTat1.1E (EATRO1125) parasites from tsetse salivary glands, our inDrop data identified both metacyclic and pre-metacyclic cells, which allowed us to evaluate the expression of *mVSG* genes before monoallelic expression was established. Second, 10xGenomics data from [9] using the RUMP503 strain were re-analyzed. Previous work concluded that each metacyclic cell expressed a single *mVSG*, as our re-analysis concluded [9]. However, expression of multiple *mVSG* genes occurs before cells become metacyclic forms. The intermediate cell cluster identified in their study therefore likely represents the pre-metacyclic cells in our re-analysis. This analysis was only possible due to the addition of *de novo* assembled *mVSG* transcripts. Third, we used smRNA-FISH in an *in vitro* metacyclogenesis model using the Lister 427 strain. Taken together, our data from three strains analyzed by different techniques, strongly support our conclusions. This triple set of experimental evidence supports a dynamic process where many genes are initially transcribed at a low level, and as expression levels increase, a single gene emerges as the active *VSG*.

Metacyclic cells express a protein coat comprised of a single VSG type [34], suggesting that the multi-<u>m</u>*VSG* gene expression we observed is restricted to the transcript level in pre-metacyclic cells. Indeed, immunogold electron microscopy [17] and immunofluorescence data [6] indicates that pre-metacyclic cells do not have VSG on their surface. According to the nomenclature from [54], VSG protein appears on the surface of nascent metacyclic cells, a developmental stage between pre-metacyclic and metacyclic form cells. As no cell division occurs between pre-metacyclic/nascent/metacyclic differentiation, these data suggest that there is a temporal component to ensuring a single VSG coat which depends upon both the time required to establish a single active locus and the differentiation from pre-metacyclic to metacyclic cell. No data is available on the duration of the transition from pre-metacyclic to metacyclic cell, so it is not possible to estimate the rate at which a single *mVSG*-ES is chosen *in vivo*. Evidence from VSG switching experiments in bloodstream forms indicates that new VSG proteins are detectable on the surface within 12 h of a switch [61], suggesting that the

establishment of monoallelic expression could occur within this time frame. During the transition from BSF to PCF following a tsetse bloodmeal, the VSG coat is removed and replaced with procyclin. The appearance of procyclin mRNA precedes the appearance of protein by approximately 6h [62]. If a similar delay exists with BARP to mVSG coat remodeling, this could potentially explain how parasites are able to express a single VSG type in their mVSG coats. Nevertheless, the exact parameters underlying these transitions remain to be determined.

In bloodstream form cells, a single *VSG* is transcribed from the active *VSG*-ES in the ESB [27]. In metacyclic cells, the picture is less clear: RNA Pol I staining indicated that no ESB is detected at this stage. Metacyclic *VSG*-ES are much shorter than BSF *VSG*-ES (~5 Kbp for *mVSG*-ES compared to ~60 Kbp for BSF *VSG*-ES), therefore the absence of an ESB in metacyclic cells may be due to fewer RNA Pol I molecules at these loci [34]. Inferring from the characteristics of the bloodstream ESB, we can speculate on nature of *VSG* gene expression regulation in metacyclic cells. The ESB is a bi-partite nuclear body consisting of the mRNA trans-splicing machinery and the actively transcribed *VSG*-ES [31]. VEX1 and VEX2 are markers of the ESB, and their association with the ESB are key to the maintenance of VSG monoallelic expression. Perturbation of either or both protein's expression leads to dramatic failures in monoallelic VSG expression control, with silent VSG-ES being transcribed and multiple VSGs expressed on the surface [29,30]. Pharmacological inhibition of transcription disperses the VEX1 and VEX2 and RNA Pol I foci [29–31,63], suggesting that the ESB requires active transcription. These data lead to a picture of the ESB as a structure which emerges from, and is stabilized by transcription of a *VSG* gene.

We propose a model for the establishment of monoallelic expression in metacyclic cells (Fig 7) where the multi-*mVSG* expression profiles observed in this study correlate to an initial step in a "race" between each *mVSG*-ES (Fig 7A) to obtain a sufficient level of transcription (Fig 7B) which leads to the down regulation of other sites. We propose that following asymmetric division of attached epimastigote parasites, pre-metacyclic cells begin transcribing multiple *mVSG*-ES. This would likely in turn recruit VEX1 and VEX2 proteins to these loci, triggering a positive feedback loop, where VEX proteins recruit the mRNA splicing machinery, stabilize the ESB and drive further transcription. This could lead to suppression of transcription at other *mVSG*-ES by depletion of these limiting factors, or by the previously proposed "RNA silencing", in which transcripts from the active *VSG* silence transcription from other *VSG*-ES [29].

We note a striking similarity between the dynamics controlling the establishment of monoallelic expression in both *mVSG* and olfactory neuron *OR* gene choice [64]. In Metazoa, monoallelic expression underpins the expression of olfactory receptors (*ORs*). It is initiated during neuronal development in the olfactory bulb, and establishment of monoallelic expression is preceded by expression of multiple olfactory receptors in early mature olfactory neurons [64]. A combination of DNA enhancer networks and an epigenetic trap, where the expression of an OR triggers a feedback loop which inhibits further activation after the first gene, establishing a single active OR. Here, during development of olfactory neurons, all *OR* genes are modified with heterochromatic histone trimethyl marks H3K4me3 and H4K20me3 [65]. The lysine demethylase LSD1 is then transiently expressed during development, activating a single OR [66]. A feedback signal triggered by OR expression utilizing the misfolded protein response inhibits further OR gene activation [67].

Both systems have intermediate developmental stages where transcripts from multiple *mVSG/OR* genes are detected, which then resolve to monoallelic states. Further, the establishment of monoallelic expression of *OR* genes is dependent on the structured and temporal control of histone methylation [65]. Similarly, in trypanosomes, the histone methyltransferase

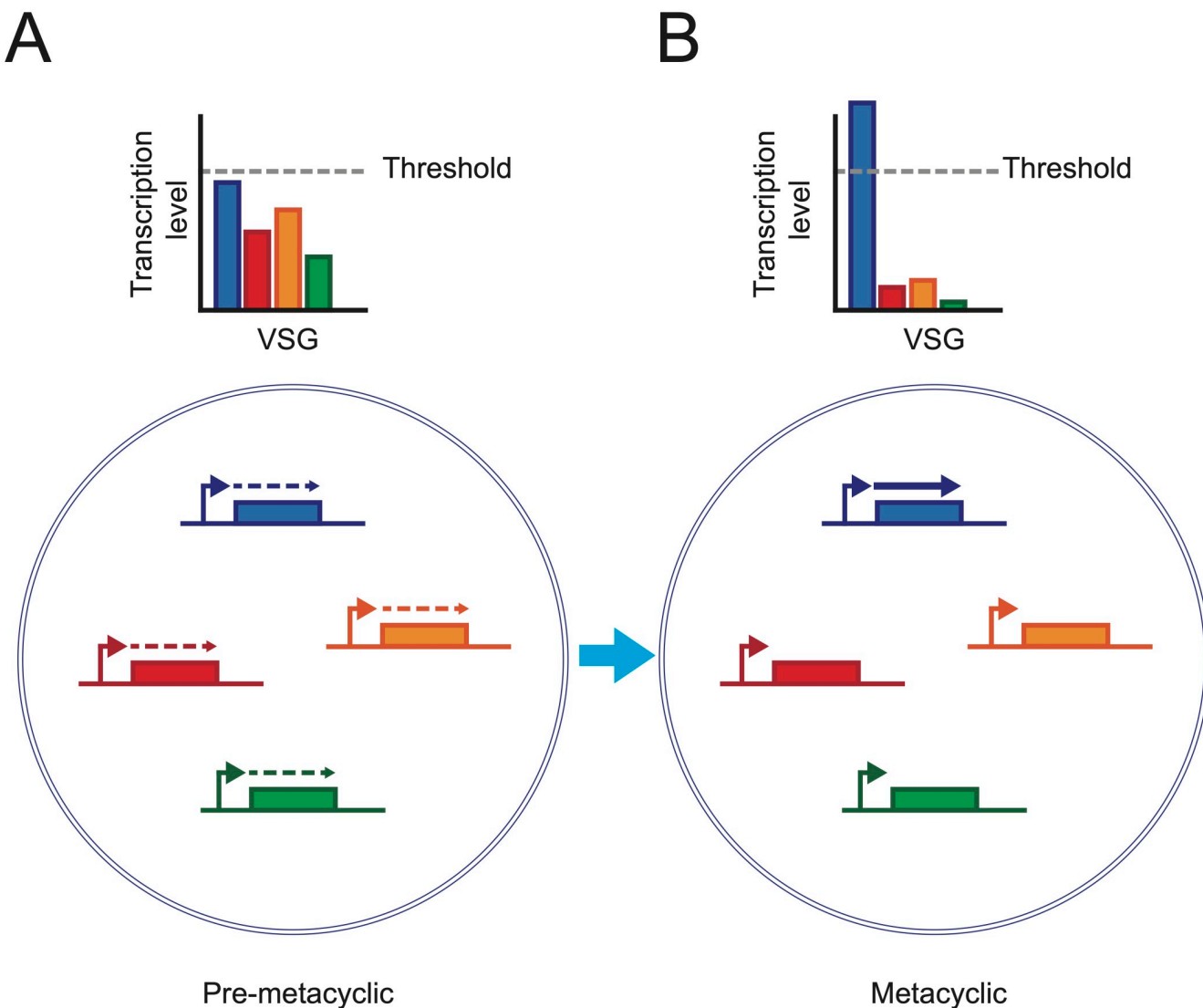

**Fig 7. Model of the initiation of *VSG* monoallelic expression during metacyclogenesis.** A. Pre-metacyclic cells initiate a transcriptional race for monoallelic expression where a single *VSG* gene must reach a threshold of transcriptional activity. B. In metacyclic cells, this threshold has been reached for a single VSG that is active, while the transcription of the other *mVSG*s is silenced.

DOT1B is required for strict control of monoallelic expression following a transcriptional switch [37]. In BSF cells, ectopic expression of a second VSG leads to silencing of the active VSG that is dependent on DOT1B [68]. In the race model, the *mVSG*-ES who lost the "race" for transcription would be silenced in a DOT1B dependent manner following the initial race, consistent with the two-stage process observed in BSF cells. These similarities suggest common solutions to the problem of monoallelic expression which deserve further investigation. Through both experimental data, and re-analyses of published data, we are able to uncover the dynamics underlying the establishment of monoallelic VSG expression. Our data provide a landscape to frame these future investigations into the establishment of single antigen expression in trypanosomes.

## Materials and methods

### Tsetse fly infections and dissections

Tsetse flies (*Glossina morsitans morsitans*) were maintained at 27˚C and 70% hygrometry in Roubaud cages, in groups of 50 male flies per cage. Teneral flies were infected with *Trypanosoma brucei* Antat1.1E (EATRO1125) during their first meal. Stumpy form trypanosomes were resuspended at $10^6$ cells per ml in SDM-79 [69] supplemented with 10% FBS (dominique Dutscher) and 60 mM N-acetylglucosamine (Peacock et al., 2006). Flies were allowed to feed on infected media through a silicone membrane. Following infection, flies were then maintained until dissection by feeding four times per week on sheep's blood in heparin. Flies were dissected as described previously [6]. All salivary glands were collected (uninfected and infected) in trypanosome dilution buffer on ice and midguts scored for the presence of trypanosomes. Salivary glands were manually broken using tweezers before being passed through a 70 μm cell strainer (ClearLine) to remove the salivary glands. Trypanosomes were counted on a hemocytometer and diluted to 9.4 x $10^4$ cells per ml. The final concentration was reached by adding 18 μL of OptiPrep Density Gradient Medium to 100 μL of cell suspension (final concentration 15% vol/vol) immediately prior to encapsulation. Cells were maintained on ice throughout this process.

### Microscopy

Live trypanosomes in SDM-79 [69] were spread on glass slides and imaged on a DMR microscope (Leica) equipped with a EMCCD camera (C-9100, Hamamatsu). The camera was controlled using μManager and a plugin for Hamamatsu cameras [70] and captured images were edited using Fiji is just ImageJ (FIJI) [71]. For the viability assay, cells were scored for motility in a hemocytometer using an inverted phase contrast microscope.

### Cell encapsulation

InDrop emulsions were prepared as described in [39]. A detailed protocol compatible with the microfluidic designs described by [39] is available on (10.5281/zenodo.3974628). Monodisperse droplets were produced by injecting the 3 aqueous phases consisting of the cells mix, RT and lysis mix and barcoded beads mix, and a fluorinated oil phase containing fluorosurfactant into a microfluidic chip with controlled flow rates, creating an emulsion. Polydimethylsiloxane (PDMS) chips were fabricated as previously described [72] at the Institut Pierre-Gilles de Gennes (IPGG) in Paris. After plasma bonding, chips were made hydrophobic by manual injection of 2% 1H,1H,2H,2H-perfluorodecyltrichlorosilane (Sigma-Aldrich) diluted in HFE 7500 (3M Novec).

Three aqueous phases, respectively for the cells, reverse transcriptase/lysis and BHMs (1CellBio) were prepared as follows; the cell mix contained 100 μL of cells diluted at a concentration of 9.4 x $10^4$ cells per ml in trypanosome dilution buffer (TDB) (5 mM KCl, 80 mM NaCl, 1 mM $MgSO_4$, 20 mM $Na_2HPO_4$, 2 mM $NaH_2PO_4$, 20 mM glucose, pH 7.4) supplemented with 18 μL Optiprep (1CellBio). The RT mix contains 1.3 RT premix (1CellBio), 11 mM $MgCl_2$ (1CellBio), 6.9 mM DTT (Invitrogen), 1.13 U/μL SUPERase IN RNAse Inhibitor (Life Technologies), 20.7 U/μL Superscript III (SSIII) (Invitrogen) and 1 μM DY-647 (Dyomics), the beads mix contains BHMs (1CellBio) packed in 1x Gel Concentration Buffer (1CellBio). The commercial BHMs carry photocleavable barcoded primers, containing, in order, a T7 promoter sequence for IVT, an Illumina adaptor, a cellular barcode, a 6 nt UMI and a 18TVN primer site, as described [39]. The closely packing of the BHBs allows 80–90% loading of a single bead per droplet [73].

The 1 ml syringes (Omnifix) for the cells, reverse transcriptase/lysis and BHMs were loaded with mineral oil (Sigma-Aldrich) and connected to a 200 μL pipette tip prefilled with mineral oil using ~30 cm of polytetrafluoroethylene (PTFE) 0.56 mm tubing and a PDMS plug connecting the tubing to the tip. Mineral oil was then ejected from the syringe until it entirely filled the pipette tip prior to aspiration of each aqueous phase. Each reagent was aspirated on ice into the 200 μL pipette tips filled with mineral oil, just before encapsulation. Cells were maintained on ice throughout the procedure by placing the pipette tip holding the cells in ice. A short length (15 cm) of PFTE 0.56 mm tubing then connected this tube to the microfluidic chip. The RT/lysis and BHB pipette tips were directly inserted into the PDMS chip, whereas the tubing carrying cells from the pipette on ice was inserted into the PDMS chip. The syringe for the oil phase was filled with carrier oil (HFE-7500 fluorinated oil (3M Novec) containing 2% w/w 008-FluoroSurfactant (RAN Biotechnologies)) and connected to the PDMS chip by PFTE 0.56 mm tubing.

Droplet production and beads loading were followed using an inverted microscope (Nikon Eclipse Ti) and High-Speed Camera (Phantom). Flow rates for droplet production were controlled with syringe pumps (neMESYS 290N Low pressure Syringe Pump, Cetoni). Droplets size and frequency of production were monitored using laser optics excitation and detection and a soluble fluorophore DY-647 present in the droplets (Dyomics) as described [72]. Flow rates of 250 μL/h, 250 μL/h, 60–80 μL/h and 400–500 μL/h for the cells, RT/lysis and BHMs and oil, respectively were maintained. A 5 cm PTFE tubing (0.3 mm inner diameter) connects the chip outlet to a 1.5 mL collection tube, prefilled with 300 μL of mineral oil. Emulsions were collected during 15 minutes on ice, corresponding to approximately 4,000 drops containing both a cell and and a BHM.

## Generation and sequencing of barcoded cDNA libraries

We made two minor alterations to the molecular biology workflow described by [39]: Our protocol omits the RNA fragmentation step following IVT of double stranded cDNA products, and we modified the primer used to reverse transcribe amplified RNA (aRNA) to make our libraries compatible with standard Illumina sequencing primers. These alterations are unlikely to affect the overall performance of inDrop, however.

### 1. Recovery of first strand cDNA

Barcoded cDNA libraries were prepared according to the protocol by [39] with small exceptions. Following encapsulation and ultra violet primer release, emulsions were incubated at 50˚C for 2h and then 70˚C for 15 minutes. Following reverse transcription (RT), emulsions were partitioned into approximately 2,000 cells per partition before breaking the emulsion by adding one drop (3μL) of 1H,1H,2H,2H-Perfluorooctanol (Sigma-Aldrich) [39]. cDNA was then stored at -80˚C until further processed.

Post-RT material was thawed on ice and 20μL of dH$_2$O (nuclease free) was added and the solution centrifuged at 16,000 g for 5 minutes at 4˚C. The aqueous phase was then applied to a 0.45 μm filter spin column (Corning CoStar Spin-X, Sigma-Aldrich) to remove the BHMs. The emulsion tube was then washed with 20μL dH$_2$O to collect as much cDNA as possible and applied to the spin column. The column was centrifuged at 16,000 g for 5 minutes at 4˚C. The emulsion tube was washed once more and supernatant collected by centrifugation once more.

### 2. Removal of unused primers

The volume of cDNA was estimated and digested by adding 100 μL of digestion mix (79 μL H$_2$O, 9 μL 10x Fast Digest Buffer (Thermo Scientific), 5 μL ExoI (20 U/μL, NEB), 7 μL HinFI

(10U/μL, Thermo Scientific)) per 70 μL of cDNA and digested at 37˚C for 30 minutes. Samples were then purified by adding 1.2x volume of room temperature AMPure XP beads (Beckman Coulter), incubating 5 minutes at room temperature, collecting the beads on a magnet and then 3 washes with freshly prepared 80% ethanol (room temperature), followed by air drying, according to the manufacturer's instructions. cDNA was then eluted in 17 μL of dH$_2$O.

### 3. Second strand synthesis

Second strand synthesis (SSS) was performed using the NEBNext Ultra II Non-Directional RNA Second Strand Synthesis Module (E6111S). 2 μL of SSS buffer and 1 μL of SSS enzyme mix was added to 17 μL of cDNA and the reaction was incubated at 16˚C for 2h30m in a PCR machine with the lid set to 16˚C. cDNA was purified again using 1.2x AMPure XP beads (as above) and eluted in 7μL of DNA elution buffer [39].

### 4. *In vitro* transcription and reverse transcription of amplified RNA (aRNA)

Double stranded cDNA was then amplified using the HiScribe T7 Quick High Yield RNA Synthesis Kit (E2050S). To the 8μL of double stranded cDNA, 10 μL of 10x NTPs buffer mix, 2 μL of T7 enzyme and 1 μL of SUPERase IN RNAse Inibitor (20 U/μL). were added and the reaction incubated at 37˚C with the lid set to 50˚C for 14–16 hours. The size and concentration of amplified RNA was verified using a BioAnalyzer RNA pico chip (Agilent) using the mRNA pico program. Approximately 5 ng of aRNA was then reverse transcribed by adding 1 μL of 10 mM dNTPs, 2 μL of Rd2N6 (see primer list) and dH$_2$O to 13 μL. The RT reaction was heated to 70˚C for 3 minutes and then put on ice for 1 minute. Finally, 4 μL of 5x first strand buffer, 1 μL of 0.1M DTT, 1 μL of SSIII (200 U/μL) and 1 μL of SUPERase IN RNase inhibitor (20U/μL). RT reactions were incubated for 5 minutes at 25˚C, followed by 1h at 50˚C and heat inactivated at 70˚C for 15 minutes. cDNA was purified by adding 20 μL of dH$_2$O then purified with 1.2 x AMPure XP beads and eluted with 20 μL dH$_2$O. 10 μL of this was then stored at -80˚C as a pre-PCR backup.

### 5. Library PCR

The remaining 10 μL of cDNA was used for the library PCR. Libraries were PCR amplified using KAPA HiFi HotStart ReadyMix (Kapa Biosystem, KK2601). 25 μL of PCR mix containing 12,5 μL of 2x KAPA HiFi Master mix, 10 μL of cDNA, together with 2.5 μL of 5 μM P5-Rd1 and P7-Rd2 primers (final concentration of 500 nM). Library indices were in the P7-Rd2 primer. Primer sequences are available in S4 Table. The libraries were amplified using the following protocol; initial denaturation 5 min at 95˚C, then 2 cycles of denaturation 98˚C / 20 s, annealing 55˚C / 30s and extension 72˚C / 40 s, followed by 10 cycles of denaturation 98˚C / 20 s, annealing 65˚C / 30 s and extension 72˚C / 40 s and final extension 5 min at 72˚C. They were then size selected using 0.7x volume of AMPure XP beads at room temperature prior to quantification and sequencing.

### 6. Library quantification and sequencing

Libraries were quantified using the KAPA library quantification kit for Illumina (Kapa Biosystem, KK4824) according to the manufacturers' protocol. Bulk libraries (Fig 1A) were sequenced on the Illumina MiSeq platform with a MiSeq V3 150 cycle cartridge at the Institut Curie. Single cell barcoding libraries were sequenced on the Illumina NextSeq platform at the Institut Pasteur using a Mid-output 150 cycle cartridge. Sequencing of single cell libraries at

BGI was performed on the MGIseq-2000 following conversion to DNBseq libraries [74]. Details of sequencing outputs including read lengths and number of reads generated can be found in S5 Table.

### Trypanosoma brucei transcriptome

We assembled a non-redundant transcriptome from the T. brucei TREU927 reference strain [26]. As only 60% of the trypanosome transcriptome has annotated 3´ UTRs, we decided to extend each gene's annotated 3´ extremity to the 5´ extremity of the neighboring gene. This was done by a combination of scripted extension using the following rules. 1. If the proceeding gene is on the same strand, and within 2.5 kbp, the gene is extended to the base before the proceeding gene. 2. If the proceeding gene is over 2.5 kbp then the gene is extended to 2.5 kbp downstream. 3. If the gene already extended into the proceeding gene, then no change was made. 4. Similarly if the proceeding gene is on the opposite strand, no extension was made. Following extension of the entire gene list, extended transcripts were filtered using a non-redundant gene list [75].

To obtain sequences of metacyclic *VSG* genes expressed in the salivary glands, we aligned our data to the VSGnome (including flanks) of the Antat1.1E (EATRO1125) strain made available by George Cross [48] here (http://129.85.245.250/index.html), using bowtie2 [76] and selected the top 8 gene most abundant transcripts. These are Tbb1125VSG-1401 (Genbank: KX699657), Tbb1125VSG-1476 (Genbank: KX699716), Tbb1125VSG-1654 (Genbank: KX699860), Tbb1125VSG-385 (Genbank: KX699226), Tbb1125VSG-393 (Genbank: KX698711), Tbb1125VSG-4564 (Genbank: KX700928), Tbb1125VSG-4862 (Genbank: KX701110), Tbb1125VSG-4959 (Genbank: KX701187).

To this non-redundant transcriptome, we appended both a human transcriptome and the *G. morsitans morsitans* genome sequence [77]. Our inDrop library preparations included 5% human B-cells as a positive control for the inDrop pipeline, and our bowtie2 index therefore contained the ENSEMBL release 85 human transcriptome [78]. The alignment index was constructed with bowtie2-build [76]. These files are available at 10.5281/zenodo.3974628.

### Generation and analysis of bulk inDrop data

Cultured procyclic form Antat1.1E (EATRO1125) trypanosomes were encapsulated using a standard inDrop microfluidic chip, except we replaced BHMs with a single primer, T7Rd2_linkerPolyTvN (S4 Table). Libraries were generated as described by [39], and as described for single cell BHMs in this manuscript, except we excluded stages relating to filtration of BHMs. Libraries were sequenced on a MiSeq at the Institut Curie (Paris) next generation sequencing platform.

Bulk data were aligned to the trypanosome genome sequence using bowtie2 [76], and resulting bam files manipulated using samtools [79]. Metagene plots and heat maps were generated using deeptools2 [80]. Gene coordinates used in the metaplot were the 5´and 3´splicing and polyadenylation signal sites (or start and stop codons if unannotated for a particular gene). Bin size was set to 10 bp and the 500 bp region downstream from each gene are unscaled. The gene list was filtered to a non-redundant set [75].

### inDrop pre-processing pipeline

To permit more usability with bespoke genome sequences (no reference sequence is available for the strain used here for tsetse infections), we generated a new inDrop data analysis pipeline for the analysis of trypanosomes, using bowtie2 and UMI-tools [76,81], rather than bowtie and the inDrop analysis pipeline [38]. Pre-processing of inDrop data was performed using the

UMI-tools package [81]. A bash script is available at (10.5281/zenodo.3974628) which performs all steps of the pre-processing. The script executes 5 steps: 1. whitelist, 2. fastqsplitter, 3. extract, 4. align and 5. count. Whitelist generates a list of acceptable barcodes, using a regular expression to pattern match the cell barcode flanked by the common "W1" sequence [38,39] permitting up to 6 errors in the W1, and requiring a string of 3 "T" nucleotides after the UMI. We set the error correction threshold for barcodes at 2, and set the expected number of cells at 2,000. We subset $10^6$ reads to perform the whitelist step (these same filtering parameters are maintained through our workflow). Fastqsplitter breaks the read files into a user-defined number of equal parts to allow downstream parallel processing. Extract then places the whitelisted cell barcodes from read 1 into the header of mRNA sequences in read 2, using the regex pattern matching parameters from whitelist. The script can initiate multiple instances of extract to reduce the time required for this stage. Align uses bowtie2 [76] to align the reads to the hybrid trypanosome, human transcriptomes and tsetse fly genome. Reads aligning to the tsetse genome were filtered out using samtools view [79]. Finally count generates count matrices at mapping quality 0 and 40. Higher mapping stringency was used for *VSG* counts to ensure accurate mapping.

## Analysis of single cell RNA-seq data

Single cell count matrices were analysed using the Seurat R package [42,44]. Data were normalized using the single cell transform algorithm [43]. Complete R scripts are available in (10.5281/zenodo.3974628) to generate all the figures in the manuscript. Count tables from mapq 0 and mapq 40 matrices were merged into a single matrix by removing VSG counts with mapq 0 and merging the mapq 40 counts to this reduced matrix. Valid barcodes were filtered by a minimum gene cut-off of 300. VSG UMI count data were extracted using fetchData, accessing the "counts" slot of the RNA assay.

To assess the sequencing depth, we employed data subsetting, as done previously [59]. Data sets were subset using samtools and UMI totals per barcode extracted in R, retaining only valid barcodes (R script on 10.5281/zenodo.3974628). To calculate the number of mapped reads per cell, a custom script was developed to count the number of aligned reads per barcode (available on 10.5281/zenodo.3974628). Subset reads were then calculated as a fraction of the total. To calculate the Levenshtein distance between valid barcodes, the barcodes for replicate 1 were extracted and all possible pairs were evaluated using a python script (10.5281/zenodo.3974628). Histogram and plotting were performed in R. To estimate the expected rate of doublets a Poisson series was generated corresponding to the predicted occupancy of cells in droplets. This was then compared to the frequency of multi-*VSG* expression observed in the pre-metacyclic cell cluster. The metabolism analysis was performed using metabolic funtion annotations and schematic for glycolysis and TCA cycle from trypanocyc [82]. Scaled (Z-scores) are from the SCT assay of Seurat in the "scaled.data" slot.

## Single molecule RNA-FISH

Overexpression of RBP6 was performed using PT1$^{RBP6-OE}$ cells (a kind gift provided by Lucy Glover, Institut Pasteur). PT1 cells are equivalent to the 2T1 BSF [83]. Wild type Lister 427 PCF cells were transfected with pHD1313 [84]. Second, a ribosomal spacer was targeted with pRPH, integrating a hygromycin resistance cassette and eGFP reporter gene (GFP tagged Sir2.3) with a RNA Pol-I promoter under the control of a tet operator [83]. The RPH locus was then disrupted with pH3E [83], replacing hygromycin with a puromycin resistance cassette, resulting in PT1 cells. Cells were induced by addition of 10 μg/ml of tetracycline to the culture medium and diluting every 24h to 2 x $10^6$ cells per ml. We made staggered inductions to obtain

3, 4 and 5 day induced cultures on the same day. We used the ViewRNA Cell Plus Assay Kit (Catalog number 88-19000-99, Thermo Fisher) for smRNA-FISH. The company designed probes based on the *VSG-397* and *VSG-531* or alpha tubulin coding sequences. We followed the manufacturer's instructions with a few minor exceptions. Before settling cells onto slides, we chilled cultures on ice and then fixed in ice cold 1% paraformaldehyde. Fixed cells were immediately washed in ice cold PBS by spinning 3x at 1,000 g for 10 minutes. Cells were settled onto 8-well glass chamber slides (Nunc Lab-Tek II Chamber Slide System) and stained according to the manufacturer's protocol. A minimum of 100 μL was used to avoid drying samples during incubations. Cells were imaged on a widefield Zeiss Axio Imager.Z2 upright microscope equipped with an LED light source, and an Axiocam 506 camera. Images were analysed using FIJI [71,85].

## Supporting information

**S1 Fig. Schematic of the life cycle of *T. brucei*, adapted from [1].** The morphology and surface protein expression of trypanosomes during differentiation is depicted in chronological order from left to right. Cell types shown here are SL: slender bloodstream form; ST: stumpy; PCF: procyclic form; Epi: epimastigote; AE: attached epimastigote; ET: epimastigote-trypomastigote dividing cell; pMT: pre-metacyclic cell; MT: metacyclic. The nucleus is depicted as a white circle / oval and the kinetoplast (mitochondrial DNA) as a yellow circle. In trypomastigote cells, the kinetoplast is posterior to the nucleus, e.g. in PCF. In epimastigote cells, the kinetoplast is anterior to the nucleus, e.g. in Epi. Cells are coloured according to the family of surface protein expressed.
(TIF)

**S2 Fig. Survival and RNA metabolism of trypanosomes expressing the indicated fusion proteins in conditions mimicking the inDrop procedure.** A. Trypanosomes collected by centrifugation and resuspended in either SDM-79 medium, or PBS and incubated at either 0°C or 27°C. Scale bar 10 μm. B. Results of live/dead assay based on trypanosome motility, by scoring live cells subjected to phase contrast microscopy. We assessed the ability of the parasites to survive starvation at 0°C for 2h in PBS. C. Histogram shows the percentage of stringently aligned reads (mapQ > 40) to the trypanosome or human transcriptomes.
(TIF)

**S3 Fig. Details of tsetse fly infections.** Flies were infected for 28 days. The table shows the characteristics of the dissection. Below is the workflow for dissections and sample collection.
(TIF)

**S4 Fig. Library construction, basic statistics and subsetting analysis.** A. Tapestation or Bioanalyzer traces for IVT and library PCR reactions to construct libraries sequenced in this study. B. Table shows the number of cells recovered, and key parameters including UMIs and genes captured. C. Subsetting analysis of the data. Reads were subsampled using samtools [79] in 10% intervals from 10% to 100% before being re-counted using UMI-tools [81]. Aligned reads per cell were counted using a custom script (10.5281/zenodo.3974628).
(TIF)

**S5 Fig. Total read counts of scRNA-seq reads aligned to Antat1.1 VSGnome [48].** Reads for SG_1 library sequenced on the Illumina NextSeq platform were aligned to the VSGnome using bowtie2 [76] and a mapping quality filter of >40 was applied.
(TIF)

**S6 Fig. Analysis of possible technical artefacts.** A. Ambient RNA analysis. Plot shows the frequency of *VSG* UMI detection in a cluster of Ramos cells. B. Histogram shows the percentage of stringently aligned reads (mapQ > 40) to the trypanosome or human transcriptomes. C. Barcode switching analysis. The plot shows the frequency of Levenshtein distances (number of changes required to permute one barcode to another) between all pairs of barcodes in replicate 1 library 1 (1,129 barcodes). Red line shows cut-off for error correction.
(TIF)

**S7 Fig. Analysis of metabolic enzymes in salivary gland clusters.** As in Fig 3B except each gene is plotted individually.
(TIF)

**S8 Fig. VSG expression analysis for attached epimastigote, pre metacyclic and metacyclic clusters.** Data presented have not been filtered (no minimum UMI cutoff per cell) and consist of 50 random cells. Units are raw (non-normalised and unscaled) UMI counts per cell. All VSG counts use a stringent mapping quality (MapQ40).
(TIF)

**S9 Fig. Plot shows the number of *VSG* genes detected per barcode versus the aligned reads per barcode stratified by cluster.** Boxplots show bounds are 25th 50th and 75th percentiles. Individual barcodes are plotted as dots.
(TIF)

**S10 Fig. Transcript abundance estimates for trinity assembled *mVSG* transcripts.** Transcripts were assembled using trinity with bulk RNA-seq data [13] and abundance estimates (TPM) were generated using Kallisto [86].
(TIF)

**S11 Fig. *mVSG* expression analysis of data from [9].** UMAP projection of scRNA-seq data indicating the classifications of clusters. B. Normalised expression values overlaid on UMAP projections for *VSG* (IlTat1.22 AJ012198.3), *BARP* (Tb927.9.15640), *HAP2* (Tb927.10.10770), *EP1-Procyclin* (Tb927.10.10260), SGM1.1 (Tb927.7.6490), ZC3H20 (Tb927.7.2660), Calflagin (Tb927.8.5470), and RBP6 (Tb927.3.2930). C. UMAP projection of scRNA-seq data overlaid with the number of *mVSG* transcripts detected per barcode. D. VSG expression profiles for each cluster. Per cluster: Left, *VSG* expression data for a random subset of 50 cells. Data are raw (unscaled) UMI counts. Each column represents a cell and each colour a different *VSG*. Right, histogram shows the number of *VSG* expressed for all cells (per *VSG* UMI count > 2, all cells in cluster). Below: table shows the percentages of cells per cluster represented on corresponding histograms.
(TIF)

**S1 Table. Cluster marker genes for midgut, attached epimastigote, gamete, pre-metacyclic and metacyclic cells.**
(XLSX)

**S2 Table. Cluster marker genes for manual sub-clustering in Fig 4.**
(XLSX)

**S3 Table. Underlying data for smRNA-FISH.** The table contains VSG expression data for each cell analysed as well as its location in the raw data files (available at 10.5281/zenodo. 3974628).
(XLSX)

**S4 Table. Primer sequences used.**
(XLSX)

**S5 Table. Details of libraries constructed and data acquired for each one.**
(XLSX)

## Acknowledgments

We thank George Cross for making the Antat1.1E (EATRO1125) VSGnome available to the research community, and for valuable discussions and comments on the manuscript. Artur Scherf and the Biomics platform at the Institut Pasteur provided access to the Illumina Next-Seq platform. Jessica Bryant and Sebastian Baumgarten provided valuable assistance with Illumina sequencing, as well as many useful discussions, and critical reading of the manuscript. Sophie Créno from the Institut Pasteur high performance computing facility helped with computational workflows. Christelle Travaillé aided with tsetse fly infections. We thank David Horn (University of Dundee) for valuable advice and discussions. We thank Lucy Glover, Serge Bonnefoy, and Aline Araujo Alves for critical reading of the manuscript. We are indebted to Eliane Thion and Lucy Glover (Institut Pasteur, Paris) for the RBP6 strain, to Alena Zíková (Institute of Parasitology, České Budějovice) for advice on inductions, and to Susanne Kramer (Universität Würzburg, Würzburg) for advice on smRNA-FISH.

## Author Contributions

**Conceptualization:** Sebastian Hutchinson, Brice Rotureau, Andrew D. Griffiths, Philippe Bastin.

**Data curation:** Sebastian Hutchinson.

**Formal analysis:** Sebastian Hutchinson, Sophie Foulon, Roberta Menafra, Brice Rotureau, Andrew D. Griffiths, Philippe Bastin.

**Funding acquisition:** Sebastian Hutchinson, Brice Rotureau, Andrew D. Griffiths, Philippe Bastin.

**Investigation:** Sebastian Hutchinson, Sophie Foulon, Aline Crouzols, Roberta Menafra.

**Methodology:** Sebastian Hutchinson, Sophie Foulon, Roberta Menafra, Brice Rotureau, Andrew D. Griffiths, Philippe Bastin.

**Project administration:** Sebastian Hutchinson, Brice Rotureau, Andrew D. Griffiths, Philippe Bastin.

**Resources:** Sebastian Hutchinson, Sophie Foulon, Roberta Menafra, Brice Rotureau, Andrew D. Griffiths, Philippe Bastin.

**Software:** Sebastian Hutchinson, Andrew D. Griffiths.

**Supervision:** Sebastian Hutchinson, Brice Rotureau, Andrew D. Griffiths, Philippe Bastin.

**Validation:** Sebastian Hutchinson, Sophie Foulon, Roberta Menafra, Brice Rotureau, Andrew D. Griffiths, Philippe Bastin.

**Visualization:** Sebastian Hutchinson.

**Writing – original draft:** Sebastian Hutchinson, Philippe Bastin.

**Writing – review & editing:** Sebastian Hutchinson, Sophie Foulon, Aline Crouzols, Roberta Menafra, Brice Rotureau, Andrew D. Griffiths.

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
