## [Decision Letter · Decision Letter 0]

12 Jul 2021

Dear Dr Hutchinson,

Thank you very much for submitting your manuscript "The establishment of variant surface glycoprotein monoallelic expression revealed by single-cell RNA-seq of Trypanosoma brucei in the tsetse fly salivary glands." for consideration at PLOS Pathogens. As with all papers reviewed by the journal, your manuscript was reviewed by members of the editorial board and by several independent reviewers. The reviewers appreciated the attention to an important topic. Based on the reviews, we are likely to accept this manuscript for publication, providing that you modify the manuscript according to the review recommendations.

Both reviewers were impressed with the analysis and appreciated the importance of the study for understanding monoallelic VSG expression. They also agreed that the study makes a significant contribution beyond the previously published single cell analysis performed with African trypanosomes. Both reviewers also offered several comments and suggestions for improving the manuscript which I encourage you to consider and address. In particular, both reviewers specifically mentioned the identified gamete cluster and ask for a more detailed description or possibly additional confirmation. Please pay particular attention to these comments. 

Sincerely,

Kirk W. Deitsch

Section Editor

PLOS Pathogens

Margaret Phillips

Section Editor

PLOS Pathogens

Kasturi Haldar

Editor-in-Chief

PLOS Pathogens

orcid.org/0000-0001-5065-158X

Michael Malim

Editor-in-Chief

PLOS Pathogens

orcid.org/0000-0002-7699-2064

Reviewer Comments (if any, and for reference):

Reviewer's Responses to Questions

**Part I - Summary**

Reviewer #1: Trypanosoma brucei has a complex life cycle, alternating between two hosts. Most of the published data are derived from bloodstream and procyclic trypanosomes, due to technical challenges of working with other fly stages.

Only recently has some light been shed on the later stages of the parasite development in the tsetse fly, but we are still far from having a clear understanding of this part of the life cycle.

Hutchinson et al. address this gap in the knowledge by applying single cell sequencing to parasites isolated from the salivary glands of the flies. Single cell sequencing is a powerful tool to dissect dynamics that so far have been only described at a population level. Overall, I think the paper is well written and the work is well designed and done.

The authors employ the right controls. They first validate the technique on well-studied trypanosome stages (bloodstream and procyclic forms), and the additional use of Ramos cells as positive control further supports their findings.

Reviewer #2: This work revisits the gene expression profile of the different development stages of T. brucei found in infected tsetse, although it focuses on the analysis of salivary gland forms. The paper is well written, the work carefully done and includes the description of a new scRNA seq methodology which has been well adapted for trypanosomes. The analyses mainly concentrates in understanding the monoallelic expression of VSG genes during trypanosome metacyclogenesis in vivo, but also touches on the changes in the metabolic pathways for energy production, also upon differentiation.

Single cell transcriptomics of trypanosome forms isolated from infected flies is not novel (Vigneron A. et al 2020 PNAS). However, in this manuscript, the authors have exploited this methodology to better understand aspects of trypanosome biology that were overlook in the Vigneron work; in fact, it makes a better (and more complete) job when looking into the regulation of the expression of mVSG genes from pre-metacyclics into metacyclics, which was then validated in vitro by FISH using RBP6 over-expressor cells. Therefore, although the work is not mechanistic and descriptive in nature, I find it significant as it increases our knowledge on the regulation of expression of T. brucei mVSGs, an information that could be also relevant for other African trypanosome species that undergo metacyclogenesis in tsetse flies. In addition, it is a good methods paper that could be expanded to other trypanosome forms or kinetoplastid parasites.

**Part II – Major Issues: Key Experiments Required for Acceptance**

Reviewer #1: The main concern is that this approach has already been used last year by Vigneron et al., who successfully sequenced the transcriptome of trypanosomes isolated from the salivary glands at the single cell level. However, I believe that the work of Hutchinson and colleagues brings some interesting additions to the previously published data.

In particular, the authors attention is focused on the expression of metacyclic VSGs before the parasites are transmitted to the mammalian host. The establishment of monoallelic expression in variant multigene families is still poorly understood in several organisms.

This work demonstrates for the first time the presence of an intermediate state in which trypanosomes express several mVSGs at low level before committing to the expression of one single VSG. The previous publication was missing this because of the lack of sequenced genome for the strain used in the experiments.

The authors do multiple validations of their observation, by reanalyzing the previously published data with newly assembled VSGs and by smFISH. From my point of view, the use of biological replicates (instead of technical) would have improved the confidence in the results, confirming that this observation is consistent across samples and is reproducible.

Nevertheless, this finding is very interesting and, in my opinion, it brings enough novelty to justify publication, despite the already available single cell data.

Another interesting new observation compared to the previously published work is the identification of a gamete cluster. Why do the authors think this cluster has not been identified in the previous work? Have they tried to re-analyze those data in order to check for HAP2 or other gametocytogenesis markers expression? The discussion part should include a comment on this aspect. Especially in this instance, employing a biological replicate would help the impact of the findings.

Reviewer #2: 1) It is intriguing that the cluster of "gamete" cells shows an increase in the expression of EP1 over BARP transcripts. Given that there are no known surface markers for "gamete" cells, it would be interesting to stain these cells with anti-EP procyclin antibodies and also anti-BARP, using the BARP staining of epis as positive control.

2) Although I'm not requesting this, I wonder if the authors tried to identify what VSG proteins are expressed by metacyclics at 28 dpi using either proteomics or IF with antibodies specific for the main VSG isoforms detected by scRNA seq. If available, this will better the paper and further complement the VSG expression analysis, but at the protein level.

**Part III – Minor Issues: Editorial and Data Presentation Modifications**

Reviewer #1: (No Response)

Reviewer #2: .Introduction: the section describing the route trypanosomes take in the tsetse gut is incorrect (ln 53-55). Parasites colonise the "anterior" not the "posterior" midgut. Also, procyclics use the "ectoperitrophic space" to migrate from the midgut to the PV not the "alimentary tract". Last, formation of epimastigotes only happens within a specific window of time in the PV; i.e. most of cells in the PV don't differentiate into epis. Please amend all these accordingly.

.Please specify which of the EP1 isoforms (EP1-1 or EP1-2) is highly expressed in salivary gland forms.

.Discussion:

The relevance of the metabolic switching during pre- to metacyclics is poorly discussed. e.g. implications for the type of carbon source used for ATP production and pre-adaptation for transmission to the vertebrate host?

Although a hypothesis, the suggestion that VEX1 and VEX2 control monoallelic expression of VSG in metacyclics is highly speculative. Is there any evidence that these proteins are indeed expressed in pre- or metacyclic forms?

PLOS authors have the option to publish the peer review history of their article (what does this mean?). If published, this will include your full peer review and any attached files.

Reviewer #1: **Yes: **Francesca Florini

Reviewer #2: **Yes: **Alvaro Acosta Serrano

Figure Files:

Data Requirements:

Reproducibility:

References:

---

## [Editor Report · Decision Letter 1]

17 Aug 2021

Dear Dr Hutchinson,

We are pleased to inform you that your manuscript 'The establishment of variant surface glycoprotein monoallelic expression revealed by single-cell RNA-seq of Trypanosoma brucei in the tsetse fly salivary glands.' has been provisionally accepted for publication in PLOS Pathogens.

Best regards,

Kirk W. Deitsch

Section Editor

PLOS Pathogens

Margaret Phillips

Section Editor

PLOS Pathogens

Kasturi Haldar

Editor-in-Chief

PLOS Pathogens

orcid.org/0000-0001-5065-158X

Michael Malim

Editor-in-Chief

PLOS Pathogens

orcid.org/0000-0002-7699-2064
---

## [Editor Report · Acceptance letter]

16 Sep 2021

Dear Dr Hutchinson,

We are delighted to inform you that your manuscript, "The establishment of variant surface glycoprotein monoallelic expression revealed by single-cell RNA-seq of Trypanosoma brucei in the tsetse fly salivary glands.," has been formally accepted for publication in PLOS Pathogens.

Best regards,

Kasturi Haldar

Editor-in-Chief

PLOS Pathogens

orcid.org/0000-0001-5065-158X

Michael Malim

Editor-in-Chief

PLOS Pathogens

orcid.org/0000-0002-7699-2064